# The ULK1-FBXW5-SEC23B nexus controls autophagy

Yeon-Tae Jeong[1,2], Daniele Simoneschi[1,2], Sarah Keegan[1,2,3], David Melville[4], Natalia S Adler[5,6], Anita Saraf[7], Laurence Florens[7], Michael P Washburn[7,8], Claudio N Cavasotto[5,6], David Fenyö[1,2,3], Ana Maria Cuervo[9], Mario Rossi[1,5,2]*, Michele Pagano[1,2,10]*

[1]Department of Biochemistry and Molecular Pharmacology, NYU School of Medicine, New York, United States; [2]Perlmutter NYU Cancer Center, NYU School of Medicine, New York, United States; [3]Institute for System Genetics, NYU School of Medicine, New York, United States; [4]Department of Molecular and Cellular Biology, Howard Hughes Medical Institute, University of California, Berkeley, Berkeley, United States; [5]Instituto de Investigación en Biomedicina de Buenos Aires (IBioBA), CONICET-Partner Institute of the Max Planck Society, Buenos Aires, Argentina; [6]Translational Medicine Research Institute (IIMT), CONICET, Facultad de Ciencias Biomédicas and Facultad deIngeniería, Universidad Austral, Pilar-Derqui, Argentina; [7]The Stowers Institute for Medical Research, Kansas, United States; [8]Department of Pathology and Laboratory Medicine, The University of Kansas Medical Center, Kansas, United States; [9]Department of Developmental and Molecular Biology, Institute for Aging Studies, Albert Einstein College of Medicine, Bronx, United States; [10]Howard Hughes Medical Institute, New York University School of Medicine, New York, United States

*For correspondence:
mrossi@ibioba-mpsp-conicet.gov.ar (MR);
michele.pagano@nyumc.org (MP)

Competing interests: The authors declare that no competing interests exist.

**Abstract** In response to nutrient deprivation, the cell mobilizes an extensive amount of membrane to form and grow the autophagosome, allowing the progression of autophagy. By providing membranes and stimulating LC3 lipidation, COPII (Coat Protein Complex II) promotes autophagosome biogenesis. Here, we show that the F-box protein FBXW5 targets SEC23B, a component of COPII, for proteasomal degradation and that this event limits the autophagic flux in the presence of nutrients. In response to starvation, ULK1 phosphorylates SEC23B on Serine 186, preventing the interaction of SEC23B with FBXW5 and, therefore, inhibiting SEC23B degradation. Phosphorylated and stabilized SEC23B associates with SEC24A and SEC24B, but not SEC24C and SEC24D, and they re-localize to the ER-Golgi intermediate compartment, promoting autophagic flux. We propose that, in the presence of nutrients, FBXW5 limits COPII-mediated autophagosome biogenesis. Inhibition of this event by ULK1 ensures efficient execution of the autophagic cascade in response to nutrient starvation.

DOI: https://doi.org/10.7554/eLife.42253.001

## Introduction

Macro-autophagy (more commonly referred to as autophagy) is a highly conserved process present in all eukaryotes, which allows the degradation of proteins and organelles by lysosomes (*Hurley and Young, 2017*; *Klionsky et al., 2016*; *Lamb et al., 2013*). It is characterized by the formation of the double-membraned autophagosome that transports cytoplasmic cargos to lysosomes, where the autophagic cargo is subjected to degradation. In a simplistic way, autophagy can be classified into 'basal' and 'induced.' The former is used to maintain cellular homeostasis by promoting the turnover

of cytoplasmic components, and the latter is part of the cellular response to stressors (e.g., to produce amino acids upon nutrient deprivation). Due to its role in many cellular processes, it is not surprising that deregulation of autophagy plays a role in many human diseases, such as neurodegenerative disorders, cancer, and infection (*Jiang and Mizushima, 2014*; *Rybstein et al., 2018*).

The UNC51-like kinase 1 [ULK1, a.k.a. autophagy-related (ATG) protein ATG1] is a master regulator of autophagy (*Dankert et al., 2016*; *Hurley and Young, 2017*; *Lamb et al., 2013*; *Mizushima, 2010*). Specifically, in response to nutrient starvation or mTOR inhibition, ULK1 is activated and, in turn, this leads to enhanced activity of the autophagy-specific class III phosphoinositide 3-kinase (PI3KC3) complex, which is comprised of VPS34 (a.k.a. PIK3C3), p150 (a.k.a. VPS15 and PIK3R4), BECLIN-1 (a.k.a. ATG6), and ATG14 (*Hurley and Young, 2017*; *Lamb et al., 2013*). The activation of the PI3KC3 complex results in the production of PI(3)P (phosphatidylinositol 3-phosphate), which is necessary for the recruitment of downstream effectors and the subsequent nucleation of the autophagosome. Next, two consecutive ubiquitylation-like reactions catalyzed by certain ATG proteins mediate the attachment of phosphatidylethanolamine to LC3 family proteins (commonly referred to as LC3 lipidation), promoting the expansion and closure of the autophagosome (*Mizushima et al., 2011*). Originally, it was thought that the autophagosome only derives from mobile, cytoplasmic vesicles that are characterized by the transmembrane protein ATG9 and that are recruited to the ER. However, it has become clear that, in response to starvation, additional sources of membrane are necessary for the formation and growth of the autophagosome (*Davis et al., 2017*; *Hurley and Young, 2017*; *Wang et al., 2014*).

The coat protein complex I I (COPII) is a multi-subunit protein complex essential for the transport of cellular cargos from the ER to the Golgi apparatus (*Fromme et al., 2008*; *Zanetti et al., 2011*). A key component of COPII is SEC23, whose importance in maintaining cellular homeostasis is highlighted by the fact that mutations in the two SEC23 paralogs (SEC23A and SEC23B) cause the human genetic diseases cranio-lenticulo-sutural dysplasia and congenital dyserythropoietic anemia type II, respectively (*Boyadjiev et al., 2006*; *Lang et al., 2006*; *Schwarz et al., 2009*). The other components of COPII are SEC13, SEC24, SEC31, and SAR1. COPII vesicles emerge from specialized domains of the ER called ER exit sites (ERES) (*Zanetti et al., 2011*). However, in response to starvation, when the secretory pathway is inhibited (*Wang et al., 2014*; *Zacharogianni et al., 2014*; *Zacharogianni et al., 2011*) and there is an urgent need for membranes to form and grow autophagosomes, ERES enlarge and patch along the ER-Golgi intermediate compartment (ERGIC) to function in autophagosome biogenesis (*Davis et al., 2017*; *Egan et al., 2015*; *Ge et al., 2017*; *Ge et al., 2014*; *Hurley and Young, 2017*; *Sanchez-Wandelmer et al., 2015*). In mammals, disruption of ERES inhibits autophagosome biogenesis at an early stage (*Stadel et al., 2015*; *Zoppino et al., 2010*). Moreover, in response to nutrient starvation, the PI3KC3 complex, which is activated by ULK1, promotes the recruitment of COPII components to the ERGIC (*Egan et al., 2015*; *Ge et al., 2013*; *Ge et al., 2014*; *Karanasios et al., 2016*). Next, specialized COPII vesicles budding from the ERGIC act as precursors for LC3 lipidation, a critical step in autophagosome biogenesis (*Egan et al., 2015*; *Ge et al., 2017*; *Ge et al., 2014*). The functions of COPII in the autophagic pathway are conserved along evolution. In fact, in response to starvation, yeast COPII components physically interact with core elements required for autophagy, and COPII vesicles provide membrane sources for the growing autophagosome (*Davis et al., 2016*; *Ishihara et al., 2001*; *Lemus et al., 2016*; *Reggiori et al., 2004*; *Tan et al., 2013*). However, how the components of COPII vesicles are regulated in response to nutrient deprivation to allow their contribution to autophagosome biogenesis is largely unknown.

SKP1-CUL1-F-box protein (SCF) complexes form a family of multi-subunit ubiquitin ligases, which, in turn, is part of the super-family of Cullin-Ring Ligase (CRL) complexes (*Petroski and Deshaies, 2005*; *Skaar et al., 2013*; *Skaar et al., 2014*). In human, 69 F-box proteins act as the substrate receptor subunits of SCF ubiquitin ligases, allowing the regulation of hundreds of substrate proteins. Thus, SCFs control a multitude of cellular processes whose deregulation is implicated in many pathologies, including cancer, neurodegenerative disorders, metabolic diseases, *etc.* (*Frescas and Pagano, 2008*; *Wang et al., 2014*). We had previously shown that, by promoting the activation of the PI3K-AKT-mTOR signalling cascade, the F-box protein FBXL2 inhibits autophagy (*Kuchay et al., 2013*). A targeted siRNA screen to deplete in U2OS cells 184 substrate receptors of human CRL complexes identified FBXW5 as a top hit involved in limiting autophagy in the presence of nutrients

(YTJ and MP, unpublished results). Thus, we studied the role of FBXW5 in regulating autophagy as described herein.

## Results

### FBXW5 binds free SEC23B to promote its ubiquitylation and proteasomal degradation

To identify SCF$^{FBXW5}$ substrates, Streptag-FLAG (SF)-tagged FBXW5 was transiently expressed in HEK293T cells and affinity purified for analysis by Multidimensional Protein Identification Technology (MudPIT) (*Florens and Washburn, 2006*; *Jeong et al., 2013*). MudPIT revealed the presence of peptides corresponding to SKP1 and CUL1 (as expected), as well as 15 unique peptides derived from the COPII coat subunit SEC23B (http://www.stowers.org/research/publications/libpb-1118). To confirm the binding between SEC23B and FBXW5 and its specificity, we screened a panel of nine human F-box proteins. SF-tagged F-box proteins were expressed in HEK293T cells and affinity precipitated to evaluate their interaction with SEC23B. We found that FBXW5 was the only F-box protein capable of co-precipitating with endogenous SEC23B (*Figure 1A*).

Next, we investigated whether SEC23B is targeted for proteolysis by FBXW5. Expression of wild-type FBXW5 resulted in a reduction in the levels of both endogenous and exogenous SEC23B, as detected by immunoblotting or immunofluorescence microscopy (*Figure 1B*, *Figure 1—figure supplement 1A–B*, and *Figure 1F*). This reduction depended on the ability of FBXW5 to form an active SCF complex as demonstrated by the observation that the expression of FBXW5(ΔF), a mutant in which the F-box domain was deleted, did not affect SEC23B protein levels (*Figure 1B*). Moreover, either co-expression with dominant negative (DN)-CUL1 (a mutant of CUL1 lacking its C-terminus, which retains the binding to F-box proteins and SKP1, but not to the catalytic subunit RBX1) or addition of MLN4924 (a NEDD8 activating enzyme inhibitor used to inhibit the activity of SCFs) blocked the FBXW5-dependent degradation of SEC23B (*Figure 1C* and *Figure 1F*). In contrast, although it has been reported that FBXW5 can form a functional CRL4$^{FBXW5}$ complex (*Hu et al., 2008*; *Kim et al., 2013*), co-expression of a CUL4 dominant negative mutant (DN-CUL4) did not interfere with the FBXW5-dependent degradation of SEC23B (*Figure 1C*). The observed reduction of SEC23B protein levels upon FBXW5 transfection was also rescued by the addition of the proteasome inhibitor MG132 (*Figure 1C*), indicating that the decrease in SEC23B levels was due to proteasome-mediated proteolysis.

To confirm that FBXW5 regulates the degradation of SEC23B, we used RNA interference to reduce FBXW5 expression. Depleting FBXW5 using three different siRNA oligos (each individually) induced an increase in both the steady state-levels and the stability of endogenous SEC23B in two different cell types (*Figure 1D* and *Figure 1—figure supplement 1C*). Moreover, expression of FBXW5, but not FBXW5(ΔF-box), promoted the in vivo ubiquitylation of SEC23B (*Figure 1E*). This was observed in HEK293T by expressing a FLAG-tagged, trypsin-resistant tandem ubiquitin-binding entity (TR-TUBE), which directly binds poly-ubiquitin chains and protects them from proteasome-mediated degradation (*Dankert et al., 2016*; *Yoshida et al., 2015*). After immunoprecipitation of FLAG-tagged TR-TUBE, high molecular weight ubiquitylated species of SEC23B were detected in lysates of cells expressing FBXW5, but not FBXW5(ΔF-box) (*Figure 1E*).

Altogether, these results indicate that FBXW5 controls the ubiquitin- and proteasome-mediated degradation of SEC23B. Intriguingly, SEC23A did not bind FBXW5, and its levels were not affected by FBXW5 silencing (*Figure 1F* and *Figure 1—figure supplement 1D–E*), indicating that SEC23B, but not SEC23A, is a substrate of SCF$^{FBXW5}$.

Immunopurified FBXW5 does not co-precipitate other COPII subunits (*Figure 1F* and http://www.stowers.org/research/publications/libpb-1118), suggesting that the subpopulation of SEC23B interacting with FBXW5 is not integrated within COPII vesicles. SEC23B and SEC24 proteins form tight heterodimers (*Fromme et al., 2008*); therefore, we reasoned that FBXW5 and SEC24 proteins might compete for the binding to SEC23B. To examine this possibility, we co-transfected HEK293T cells with FBXW5 and increasing amounts of SEC24B and subjected the resulting lysates to affinity purification. In line with our hypothesis, increasing concentrations of SEC24B resulted in a marked decrease in the binding of endogenous SEC23B to FBXW5 (*Figure 1G*). Increasing amounts of SEC24B also induced an increase in the levels of endogenous SEC23B (*Figure 1F*), likely due to the

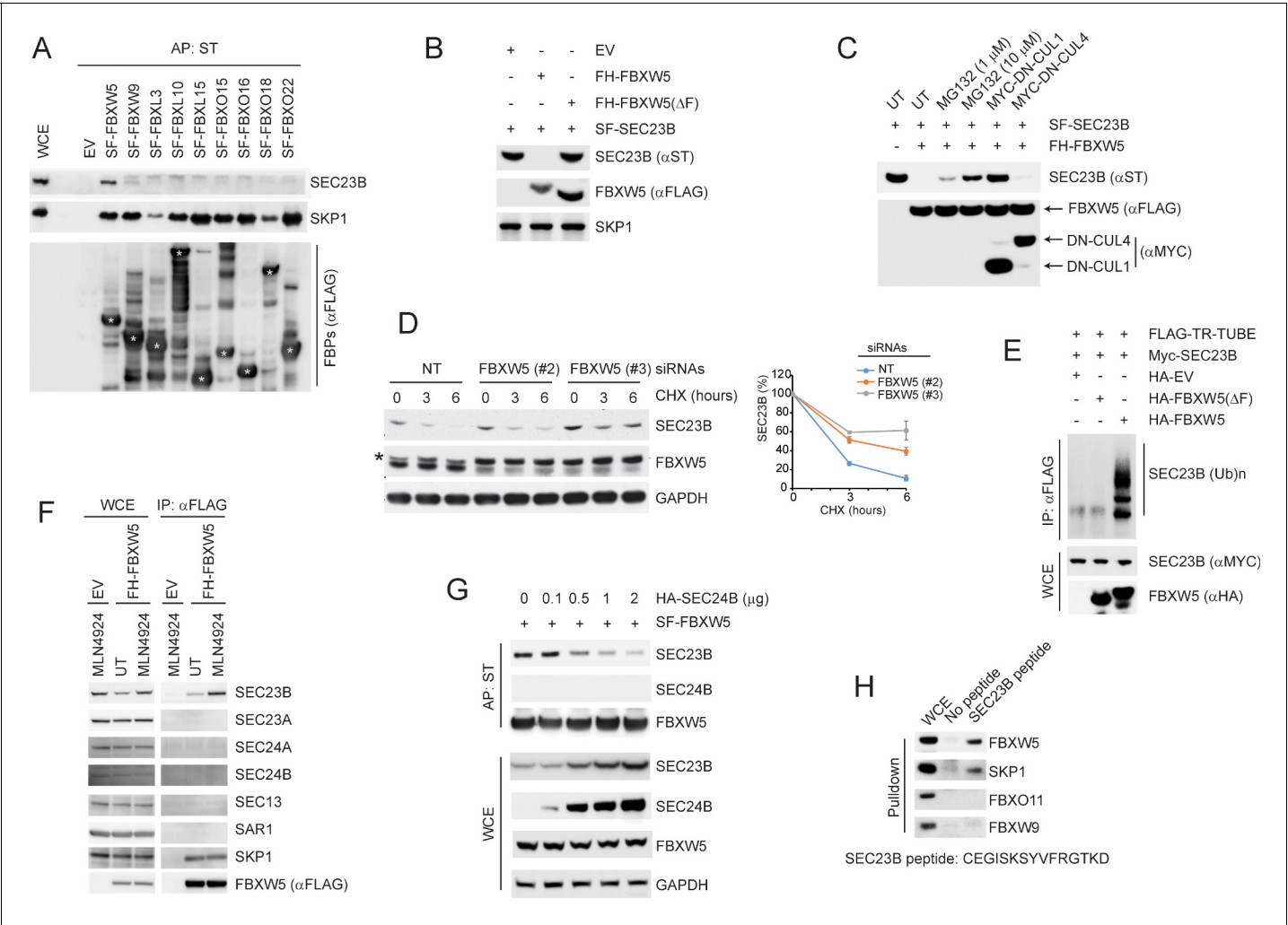

**Figure 1.** SCF^FBXW5 interacts with SEC23B and targets it for ubiquitylation and proteasome-mediated degradation. (**A**) HEK293T cells were transfected with either an empty vector (EV) or the indicated Streptag-FLAG-tagged (SF) F-box proteins (FBPs). Twenty-four hours after transfection, cells were treated with MLN4924 for 4 hr before harvesting them for affinity-purification (AP) with Streptactin (ST) beads and immunoblotting as indicated. (WCE, whole cell extracts). The white asterisks indicate individual F-box proteins. (**B**) HEK293T cells were transfected with an EV, FLAG-HA-tagged FBXW5 (FH-FBXW5), or FH-FBXW5(ΔF) together with SF-tagged SEC23B. Twenty-four hours after transfection, cells were harvested for immunoblotting. (**C**) HEK293T cells were transfected with FH-FBXW5 and SF-SEC23B in combination with either an EV, MYC-tagged DN-CUL1, or MYC-tagged DN-CUL4 as indicated. Twenty-four hours after transfection, cells were either left untreated (UT) or treated with MG132 for 6 hr, and finally harvested for immunoblotting. (**D**) U-2OS cells were transfected with either a non-targeting siRNA oligo (NT) or two different FBXW5 siRNA oligos (individually). Seventy-two hours after siRNA transfection, cells were treated with cycloheximide (CHX) for the indicated times and harvested for immunoblotting. The asterisk indicates a nonspecific band. The graph shows the quantification of SEC23B levels from three independent experiments. Error bars indicate standard deviation. (**E**) HEK293T cells were transfected with an EV, HA-tagged FBXW5, or HA-tagged FBXW5(ΔF) together with MYC-tagged SEC23B and FLAG-TR-TUBE cDNA as indicated. WCLs were immunoprecipitated (IP) with anti-FLAG resin and immunoblotted as indicated. The line on the right marks a ladder of bands corresponding to poly-ubiquitylated SEC23B. (**F**) HEK293T cells were transfected with either an EV or FH-FBXW5. Twenty-four hours after transfection, cells were either left untreated (UT) or treated with MLN4924 for 4 hr before harvesting them for immunoprecipitation (IP) with FLAG beads and immunoblotting as indicated. (**G**) HEK293T cells were transfected with SF-FBXW5 and increasing amounts of HA-tagged SEC24B as indicated. Twenty-four hours after transfection, cells were harvested for affinity-purification (AP) with Streptactin (ST) beads and immunoblotting as indicated. (**H**) WCEs from HEK293T cells were incubated with either unconjugated beads or beads coupled to a SEC23B peptide (a.a. 180–194, CEGISKSYVFRGTKD). Beads were washed with lysis buffer and bound proteins were eluted and subjected to SDS-PAGE and immunoblotting.
DOI: https://doi.org/10.7554/eLife.42253.002

The following source data and figure supplement are available for figure 1:

**Source data 1.** Source data for *Figure 1B*.
DOI: https://doi.org/10.7554/eLife.42253.004
**Figure supplement 1.** FBXW5 interacts with and promotes the degradation of SEC23B.
*Figure 1 continued on next page*

*Figure 1 continued*

DOI: https://doi.org/10.7554/eLife.42253.003

ability of SEC24B to impair the interaction between SEC23B and FBXW5. These results support the hypothesis that SEC23B cannot simultaneously bind to FBXW5 and SEC24B, suggesting that these two proteins compete for the same binding region on SEC23B. In agreement with this model, using a panel of SEC23B deletion mutants, we found that the binding to FBXW5 is mediated by the TRUNK domain of SEC23B (*Figure 1—figure supplement 1F–H*), which has been shown to mediate the SEC23-SEC24 interaction (*Mancias and Goldberg, 2008*). In fact, SEC23B(100-767), a deletion mutant containing the TRUNK domain, was able to co-precipitate endogenous FBXW5, whereas SEC23B(400-767), a mutant missing the TRUNK domain, was not (*Figure 1—figure supplement 1F–H*).

Next, we used an immobilized synthetic peptide containing the previously reported SEC24-interacting sequence of SEC23B (a.a. 180–194 in human, see *Figure 1—figure supplement 1F* and *Figure 1—figure supplement 1H*) (*Mancias and Goldberg, 2008*) and tested its ability to bind endogenous FBXW5 present in cellular extracts. While the immobilized SEC23B peptide efficiently bound FBXW5, it failed to pull down other F-box proteins (*Figure 1H*), suggesting that this 14-amino acid region is sufficient for the binding to FBXW5.

Taken together, these results indicate that FBXW5 and SEC24B associate with SEC23B in a mutually exclusive fashion and may compete for the same binding region in SEC23B.

## ULK1 phosphorylates SEC23B on serine 186 that is present in the binding motif for FBXW5 and SEC24

The interactions between F-box proteins and their cognate substrates are often regulated by post-translational modifications (most often phosphorylation) (*Skaar et al., 2013*). Therefore, we looked for phosphorylation consensus sequences for known kinases within the FBXW5-binding region of SEC23B and found that Serine 186 (S186) is part of a highly conserved ULK1 phosphorylation motif (*Egan et al., 2015*) (*Figure 2—figure supplement 1A–B*). We first investigated whether S186 is phosphorylated in cells. To this end, we generated a phospho-specific antibody against a peptide (a.a. 180–194) containing phosphorylated Serine at position 186. This antibody specifically detected the phosphopeptide, but not the unphosphorylated peptide (*Figure 2—figure supplement 1C*). Moreover, it recognized both wild-type SEC23B and a SEC23B(S186D) mutant (which mimics Ser186 phosphorylation), but not the SEC23B(S186A) mutant (*Figure 2—figure supplement 1D*), providing evidence that SEC23B is phosphorylated in vivo on S186.

Next, to test whether Ser186 on SEC23B can serve as a phospho-acceptor for ULK1, we co-expressed FLAG-tagged, wild-type SEC23B together with either wild type ULK1 or a kinase-dead (KD) mutant. Only wild-type ULK1 induced the phosphorylation of both endogenous and exogenous SEC23B on S186, as detected with our phospho-specific antibody (*Figure 2A* and *Figure 2—figure supplement 1E*) (endogenous phosphorylated SEC23B was detected in whole cell extracts, and exogenous phosphorylated SEC23B was detected in both cell extracts and anti-FLAG immunoprecipitates). In contrast, in cells expressing SEC23B(S186A), only endogenous SEC23B was phosphorylated by ULK1(*Figure 2A* and *Figure 2—figure supplement 1E*). Addition of a specific ULK1 inhibitor, SBI-0206965, blocked the ULK1-dependent phosphorylation of endogenous SEC23B in a dose-dependent manner, similar to what observed for ATG13 (*Figure 2B*), a canonical substrate of ULK1. We further confirmed that SEC23B is an ULK1 substrate by using purified proteins in an in vitro phosphorylation assay (*Figure 2C*).

Next, we studied how nutrient deprivation, a condition that activates ULK1 and autophagy, modulates the phosphorylation of SEC23B on S186. After substituting the growth medium, which contains 10% fetal bovine serum, with Earle's balanced salt solution (EBSS), we observed dephosphorylation of ULK1 and phosphorylation of ATG13 and BECLIN-1, two known substrates of ULK1 (*Figure 2D–E*), indicating that mTORC1 activity is inhibited and ULK1 and autophagy are activated. Starvation also promoted the co-localization of SEC23B and ULK1 (*Figure 2—figure supplement 1F*). Moreover, EBSS induced a time-dependent increase in the phosphorylation of endogenous SEC23B similar to what observed for ATG13 and BECLIN-1, and this phosphorylation

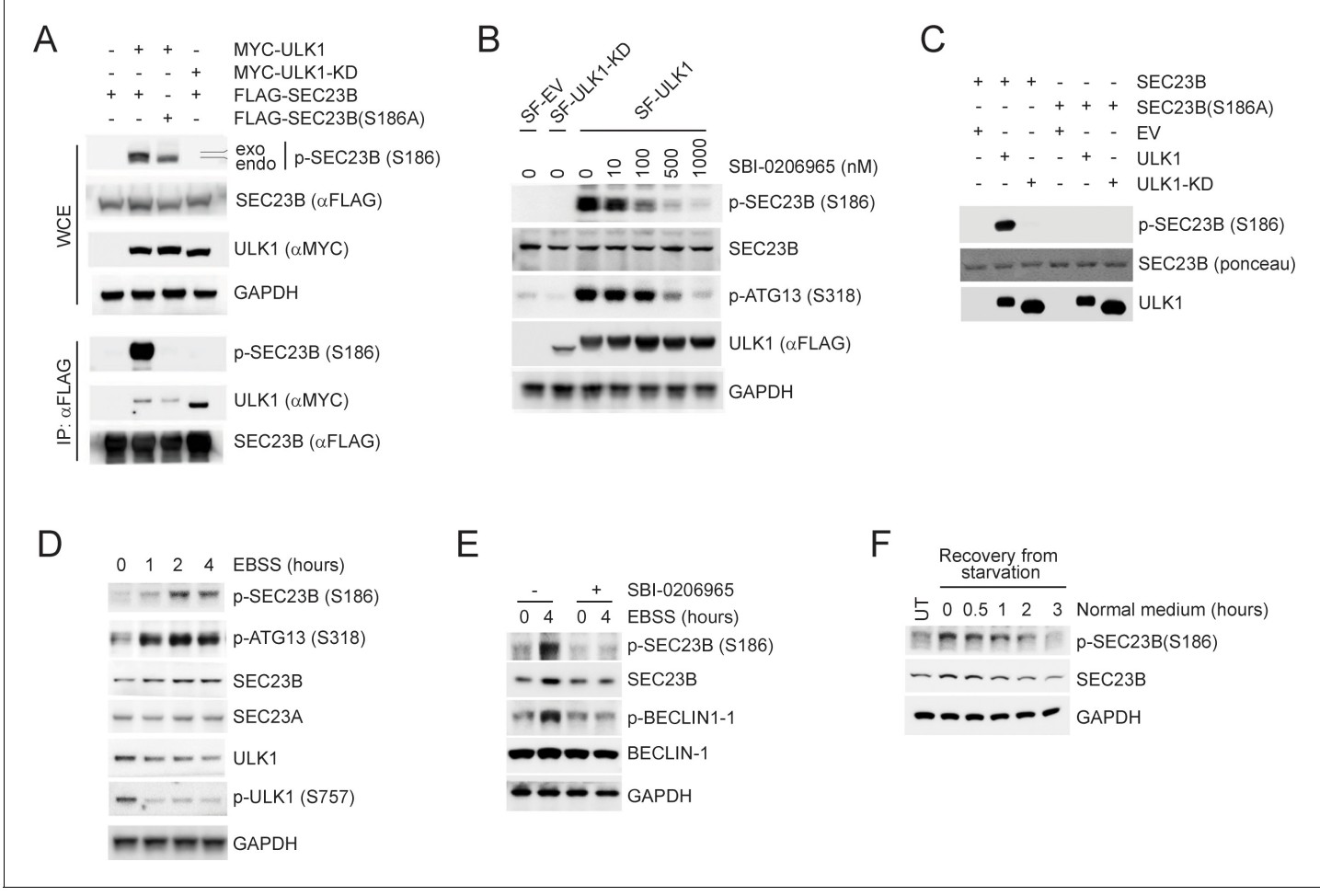

**Figure 2.** ULK1 phosphorylates SEC23B on Serine 186. (**A**) HEK293T cells were transfected with either FLAG-tagged SEC23B or FLAG-tagged SEC23B (S186A) in combination with MYC-tagged ULK1 or MYC-tagged ULK1-KD as indicated. Twenty-four hours after transfection, cells were harvested for immunoprecipitation (IP) and immunoblotting. *exo* and *endo* indicate the exogenous and endogenous SEC23B, respectively. (**B**) HEK293T cells were transfected with the SF-ULK1 or SF- ULK1-KD as indicated. Twenty-four hours after transfection, cells were treated with various doses of SBI-0206965 (an ULK1 inhibitor) for 4 hr before harvesting them for immunoblotting. (**C**) In vitro kinase assays were performed using purified SEC23B (wild-type or the S186A mutant) and ULK1 (wild-type or a kinase-dead mutant) as substrate and kinase, respectively. Purified SEC23B and ULK1 proteins were prepared by immunoprecipitation (followed by elution) from extracts of HEK293T cells transfected with each corresponding plasmid. (**D**) HEK293T cells were nutrient-starved with EBSS for the indicated times and harvested for immunoblotting. (**E**) HEK293T cells were nutrient-starved with EBSS for the indicated times (in the presence or absence of SBI-0206965) and harvested for immunoblotting at the indicated times. (**F**) HEK293T cells were recovered from nutrient-starvation (EBSS for 4 hr) for the indicated times and harvested for immunoblotting.

DOI: https://doi.org/10.7554/eLife.42253.005

The following source data and figure supplements are available for figure 2:

**Figure supplement 1.** Characterization of the phospho-SEC23B (Ser186) antibody and the phospho-mimetic SEC23B mutant.
DOI: https://doi.org/10.7554/eLife.42253.006
**Figure supplement 1—source data 1.** Source data for panel F.
DOI: https://doi.org/10.7554/eLife.42253.007

was inhibited by treating cells with SBI-0206965 (*Figure 2D–E*). Recovery from nutrient deprivation by addition of fetal bovine serum for three hours resulted in the de-phosphorylation of SEC23B and a decrease in its total levels (*Figure 2F*).

We concluded that, in response to starvation, ULK1 phosphorylates SEC23B on S186.

## ULK1-dependent phosphorylation of SEC23B inhibits its interaction with and degradation via FBXW5

To investigate whether phosphorylation on S186 affects the SEC23B-FBXW5 interaction, we used a phosphorylated version of the synthetic peptide employed for the binding experiments. While the non-phosphorylated peptide efficiently bound FBXW5, as previously observed (*Figure 1H*), a corresponding peptide containing phosphorylated Ser186 displayed a strongly reduced ability to bind FBXW5 (*Figure 3—figure supplement 1A*). This indicates that Ser186 phosphorylation inhibits the interaction between these two proteins, in agreement with the fact that the binding of FBXW5 to the phospho-mimetic SEC23B(S186D) mutant is abolished, while FBXW5 binding to SEC23B(S186A) is increased (*Figure 3A*). Accordingly, endogenous SEC23B phosphorylated on Ser186 was not affinity purified with FBXW5 (*Figure 3B*). Similarly, upon nutrient deprivation (when SE23B becomes phosphorylated on Ser186) the interaction between FBXW5 and SEC23B decreased (*Figure 3—figure supplement 1B*).

Consistent with the observed impaired interaction with FBXW5, the levels of SEC23B(S186D) did not decrease in the presence of FBXW5 (*Figure 3C*) and it exhibited a longer half-life than wild-type SEC23B (*Figure 3D*). These results suggest that ULK1 phosphorylates Ser186 in SEC23B, inhibiting its interaction with and degradation via FBXW5. Concurring with this hypothesis, silencing of ULK1, but not ULK2, reduced the levels of SEC23B phosphorylated on Ser186 and induced a decrease in the levels of endogenous SEC23B in starved cells (*Figure 3E*). Moreover, co-silencing FBXW5 rescued the decrease in SEC23B levels induced by the depletion of ULK1 but did not alter the amount of phosphorylated SEC23B (*Figure 3E*), confirming that ULK1 and FBXW5 have an antagonistic effect on the control of SEC23B protein abundance. Finally, ULK1 silencing had no effects on the levels of SEC23A (*Figure 3E*), and SEC23A levels do not increase upon starvation (*Figure 2D*).

Significantly, although Ser186 is located at the interface between the SEC23 and SEC24 heterodimer binding region (*Mancias and Goldberg, 2008*) (*Figure 1—figure supplement 1H*) and its phosphorylation blocks the SEC23B-FBXW5 interaction (*Figure 3A–B*), endogenous phosphorylated SEC23B interacted with SEC24B (*Figure 3—figure supplement 1C*). Moreover, the phospho-mimetic SEC23B(S186D) mutant interacted with SEC24A, SEC24B, SEC13, SE16, and SEC31 (*Figure 3—figure supplement 1D*, see also Figure 6A). In addition, both wild-type SEC23B and phospho-mimetic SEC23B(S186D) efficiently bound to the small GTPase SAR1(H79G), a GTP-bound SAR1 mutant that is constitutively associated with COPII vesicles, but not with the cytosolic GDP-bound SAR1(T39N) mutant (*Venditti et al., 2012*) (*Figure 3—figure supplement 1E*).

Altogether, these results indicate that the ULK1-dependent phosphorylation of SEC23B blocks its FBXW5-dependent degradation but does not interfere with either the formation of the SEC23B-SEC24 heterodimer or its recruitment to the COPII vesicle coat by SAR1.

## The FBXW5-mediated degradation of SEC23B limits the autophagic flux in the presence of nutrients

Since ULK1 plays an essential role in the induction of autophagy, and since COPII proteins, in addition to their role in secretion, are also required for the proper execution of the autophagic program (see Introduction), we investigated whether the ULK1-dependent regulation of the interaction between FBXW5 and SEC23B regulates the autophagic flux. Automated quantification of both endogenous LC3 (*Figure 4A*) and exogenous GFP-LC3 (*Figure 4—figure supplement 1A*) showed that the LC3 puncta area increased upon FBXW5 downregulation in cells grown in the presence of nutrients. Measurement of area of LC3-positive puncta was preferred to individual number of puncta per cell because clustering of the vesicular compartments in some images made delineation of individual vesicles inaccurate. The increase in LC3-positive puncta could in principle reflect either an increase in autophagic activity or an impairment in the lysosome-dependent degradation of lipidated LC3 (*Klionsky et al., 2016*). Compared to untreated cells, treatment of FBXW5-depleted cells with bafilomycin A1, a proton ATPase inhibitor that blocks the degradation of lipidated LC3 but not the formation of autophagosomes, significantly increased LC3 puncta (*Figure 4A* and *Figure 4—figure supplement 1A*), as well as the amount of lipidated LC3 (*Figure 4—figure supplement 1B*). These results suggest that depletion of FBXW5 increases autophagic flux during unstressed conditions (i.e., it increases basal autophagy). Consistent with this hypothesis, FBXW5 downregulation did not induce a further increase in the appearance of LC3 puncta upon induction of autophagy by nutrient

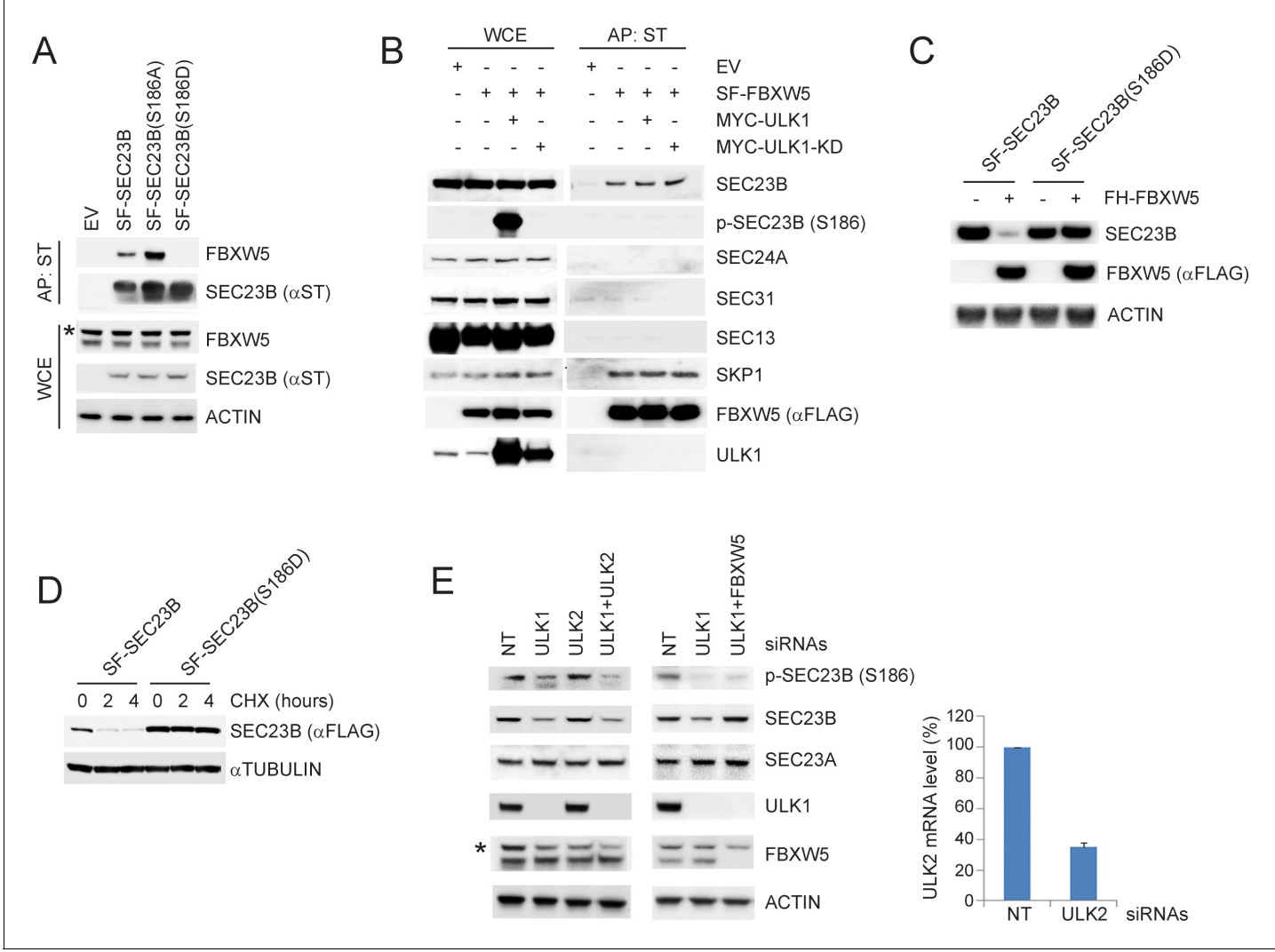

**Figure 3.** ULK1-mediated phosphorylation of SEC23B on S186 prevents the FBXW5-dependent degradation of SEC23B. (**A**) HEK293T cells were transfected with either EV, SF-SEC23B, or the indicated SF-SEC23B mutants. Twenty-four hours after transfection, cells were treated with MLN4924 for 4 hr before harvesting them for affinity-purification (AP) with Streptactin (ST) beads and immunoblotting. The asterisk indicates a nonspecific band. (**B**) HEK293T cells were transfected with SF-FBXW5 in combination with either MYC-tagged ULK1 or MYC-tagged ULK1-KD. Twenty-four hours after transfection, cells were treated with MLN4924 for 4 hr before harvesting them for affinity-purification (AP) with Streptactin (ST) beads and immunoblotting as indicated. (**C**) HEK293T cells were transfected with FH-FBXW5 in combination with SF-SEC23B or SF-SEC23B(S186D). Twenty-four hours after transfection, cells were harvested for immunoblotting. (**D**) U-2OS cells stably infected with viruses expressing either SEC23B or SEC23B (S186D) were treated with cycloheximide for the indicated times. The cells were then harvested for immunoblotting. (**E**) RPE1-hTERT cells were transfected with siRNAs against the indicated mRNAs. Sixty-eight hours after transfection, cells were nutrient-starved with EBSS for 4 hr and harvested for immunoblotting (left panel) and real-time PCR using ULK2 and GAPDH primers (right panel). The asterisk indicates the nonspecific band.

DOI: https://doi.org/10.7554/eLife.42253.008

The following source data and figure supplement are available for figure 3:

**Source data 1.** Source data for *Figure 3E*.

DOI: https://doi.org/10.7554/eLife.42253.010

**Figure supplement 1.** ULK1-mediated phosphorylation of SEC23B on S186 prevents the FBXW5-dependent degradation of SEC23B.

DOI: https://doi.org/10.7554/eLife.42253.009

starvation (*Figure 4—figure supplement 1C*), a condition under which the FBXW5-mediated degradation of SEC23B is inhibited by ULK1 (*Figure 3*). We also evaluated the turnover of p62, a cargo adaptor and substrate of autophagy (*Galluzzi et al., 2018*). Upon FBXW5 silencing, p62 degradation was activated in two different cell lines under nutrient-rich conditions (*Figure 4B*).

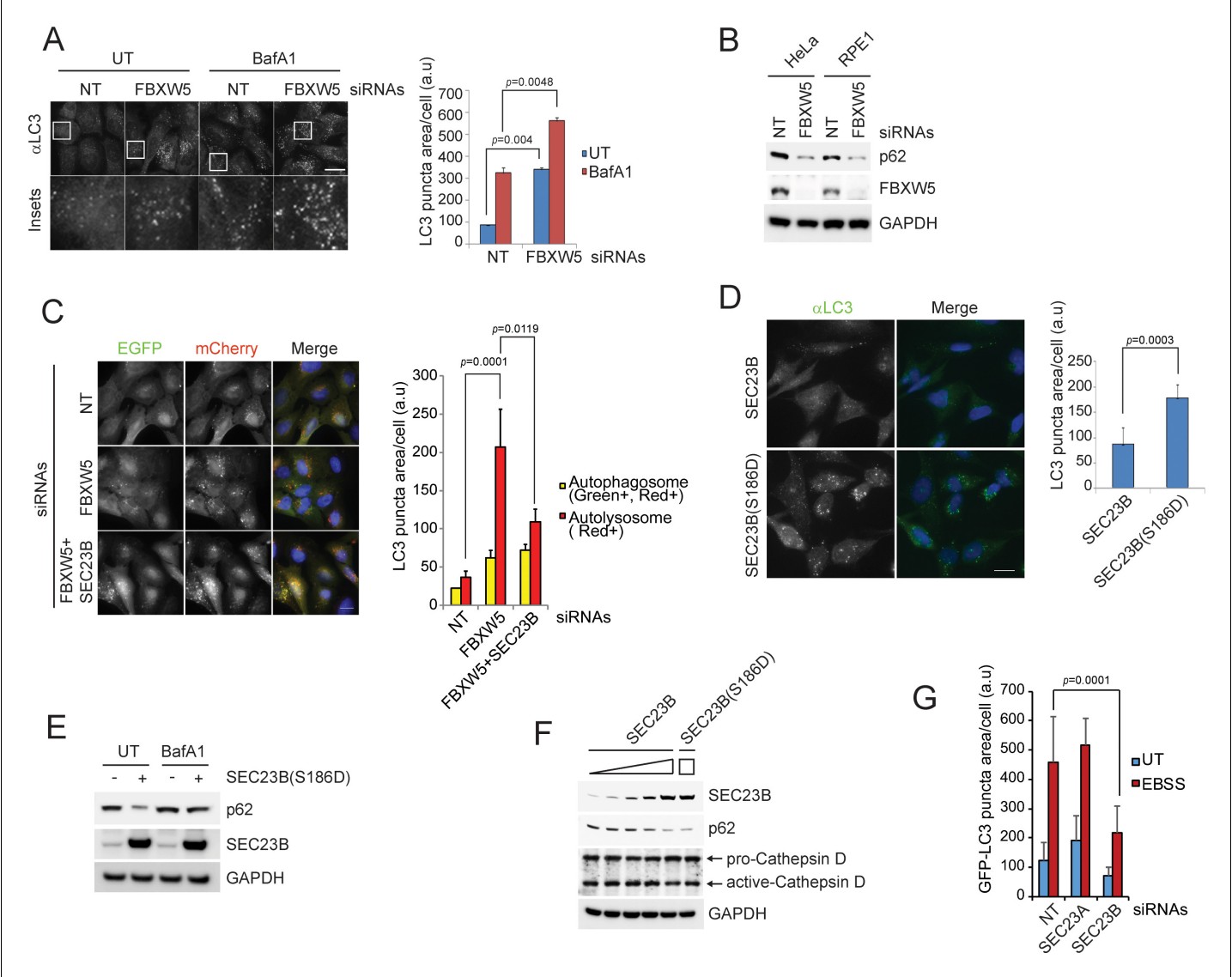

**Figure 4.** The FBXW5-mediated degradation of SEC23B limits the autophagic flux in the presence of nutrients. (**A**) RPE1-hTERT cells were transfected with a non-targeting (NT) oligo or a FBXW5-targeting siRNA oligo. Forty-eight hours after transfection, cells were re-plated onto coverglass for immunofluorescence with an anti-LC3 antibody. Where indicated, cells were treated with Bafilomycin A1 (BafA1) for 4 hr before fixation. Images of endogenous LC3 puncta underwent automated processing with at least 300 cells counted per sample. Because in several images LC3·puncta were too close to be distinguished, we adopted LC3 puncta area as a criterion for our analysis. The data are presented as mean ±SD (right panel). Scale bar, 10 μm. (**B**) HeLa and RPE1-hTERT cells were transfected with the indicated siRNAs. Seventy-two hours after transfection, the cells were harvested for immunoblotting. (**C**) U-2OS cells stably expressing tandem fluorescent-tagged LC3 (pBabe-mCherry-EGFP-LC3) were transfected with a NT oligo or a FBXW5 siRNA oligo, alone or in combination with a SEC23B-targeting siRNA oligo as indicated. Forty-eight hours after transfection, cells were replated onto coverglass, followed by fixation twenty-four hours after replating. Images of mCherry-EGFP-LC3 puncta underwent automated processing with at least 100 cells counted per sample. The data are presented as mean ±SD (right panel). The yellow and red bars represent green +red double positive LC3 puncta (autophagosome) and red only positive LC3 puncta (autolysosome), respectively. Scale bar, 10 μm. (**D**) U-2OS cells were infected with lentiviruses expressing either wild-type SEC23B or SEC23B(S186D). Twenty-four hours after infection, cells were fixed for immunofluorescence. Images of endogenous LC3 puncta underwent automated processing with at least 300 cells counted per sample. The data are presented as mean ±SD (right panel). Scale bar, 10 μm. (**E**) U-2OS cells were infected with lentivirus expressing SEC23B(S186D). Where indicated, forty-eight hours after infection, cells were treated with BafA1 prior to harvest and immunoblotting. (**F**) U-2OS cells were infected with the increasing amounts of lentivirus expressing SEC23B. Forty-eight hours after infection, cells were harvested for immunoblotting. (**G**) RPE1-hTERT cells stably expressing GFP-tagged LC3 were transfected with a NT oligo or the indicated siRNA oligos. Forty-eight hours after transfection, cells were replated onto coverglass, followed by treatment with EBSS for 1 hr and fixation. Images of GFP-LC3 puncta underwent automated processing with at least 300 cells counted per sample. The data are presented as mean ±SD.

DOI: https://doi.org/10.7554/eLife.42253.011

*Figure 4 continued on next page*

*Figure 4 continued*

The following source data and figure supplements are available for figure 4:

**Source data 1.** Source data for *Figure 4A,C,D and G*.

DOI: https://doi.org/10.7554/eLife.42253.014

**Figure supplement 1.** The FBXW5-mediated degradation of SEC23B limits the autophagic flux in the presence of nutrients.

DOI: https://doi.org/10.7554/eLife.42253.012

**Figure supplement 1—source data 1.** Source data for panels A and C.

DOI: https://doi.org/10.7554/eLife.42253.013

Next, we used a tandem fluorescent-tagged LC3 construct (mCherry-GFP-LC3) that allows monitoring autophagosome maturation as a change from double-positive, green +red (i.e., yellow) fluorescent vesicles (autophagosomes), to single-positive, red fluorescent vesicles (autolysosomes) deprived of GFP fluorescence due to its quenching at low pH (*Klionsky et al., 2016*). Using this method, we demonstrated that the majority of LC3 puncta that accumulated upon FBXW5 depletion were autolysosomes and not autophagosomes (*Figure 4C*), further suggesting that increased levels of SEC23B promotes autophagic flux.

Importantly, co-depletion of SEC23B almost completely prevented the increase in autophagy mediated by FBXW5 silencing (*Figure 4C* and *Figure 4—figure supplement 1B*). Moreover, expression of SEC23B(S186D) significantly induced higher levels of LC3 puncta compared to wild-type SEC23B (*Figure 4D*) and activated p62 degradation under nutrient-rich conditions (*Figure 4E*). We also observed that high levels of wild-type SEC23B were able to induce p62 degradation (*Figure 4F*), suggesting that a threshold of SEC23B protein expression (and not the presence of the phospho-mimetic mutation per se) is a critical determinant to induce autophagy. In agreement with a role for SEC23B in promoting autophagy, SEC23B silencing inhibited autophagy in response to starvation (*Figure 4G*). Finally, although, expression of SEC23B(S186D) is sufficient to induce autophagy in the presence of nutrients, it was unable to rescue autophagy in ULK1 knockdown cells (*Figure 4—figure supplement 1D*), likely because ULK1 is necessary to phosphorylate additional pro-autophagic substrates (*Hurley and Young, 2017*) and promote the assembly of COPII complexes (*Joo et al., 2016*).

## Ser186 in SEC23B is necessary for its localization to the ERGIC and an efficient autophagic response upon nutrient deprivation

The ERGIC compartment produces vesicles that are active for LC3 lipidation (*Egan et al., 2015*; *Ge et al., 2013*; *Ge et al., 2014*). We found that, whereas upon starvation wild-type SEC23B colocalized approximately twice more with the ERGIC membrane marker ERGIC53, the localization of SEC23B(S186A) did not change after nutrient deprivation (*Figure 5A*). Notably, SEC23B(S186D) colocalization with ERGIC53 was already high in the presence of nutrients. Accordingly, compared to wild-type SEC23B, SEC23B(S186D) co-distributed approximately twice more with ERGIC53 in fractionated membranes (*Figure 5—figure supplement 1A*) and ULK1 silencing inhibited SEC23B-ERGIC53 colocalization upon starvation (*Figure 5B*). Finally, evaluation of the secretory pathway by assaying secreted Gaussia Luciferase (*Badr et al., 2007*) indicated that overexpression of either SEC23B(S186D) or ULK1, but not wild-type SEC23B, SEC23B(S186A), or ULK1 knockdown, resulted in an inhibition of secretion in the presence of nutrients, similarly to what observed when cells were staved (*Figure 5C–D*).

Altogether, these results suggest that ULK1-mediated phosphorylation of SEC23B promotes both its localization to the ERGIC and its autophagic function, possibly at the expense of its secretory function.

Next, we used a CRISPR/Cas9-dependent strategy to generate A375 cells in which all three alleles of *SEC23B* were mutated to SEC23B(S186A) (*Figure 5—figure supplement 1B*). In contrast to wild-type SEC23B, levels of SEC23B(S186A) did not increase upon starvation (*Figure 5E*). Significantly, automated quantification of endogenous LC3 showed that, in contrast to parental cells, nutrient deprivation-induced autophagy was strongly reduced in SEC23B(S186A) knock-in cells, while basal autophagy remained unperturbed (*Figure 5F*). Finally, in response to starvation, p62 was degraded in parental cells but not in SEC23B(S186A) knock-in cells (*Figure 5E*).

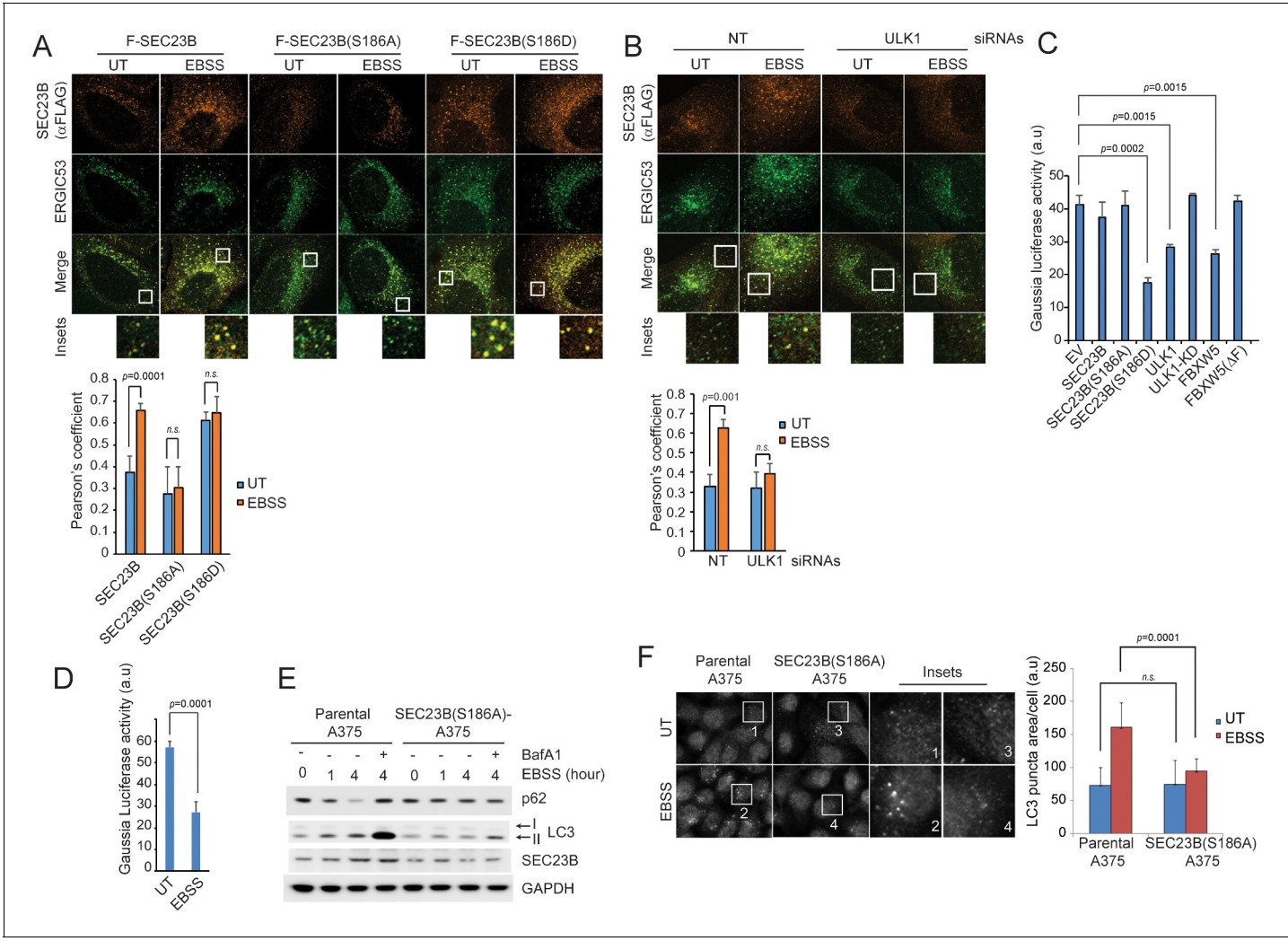

**Figure 5.** Ser186 in SEC23B is necessary for its localization to the ERGIC and an efficient autophagic response upon nutrient deprivation. (A) U-2OS cells were transfected with either FLAG-HA-tagged wild-type SEC23B, SEC23B(S186A), or SEC23B(S186D). Twenty-four hours after transfection, cells were fixed for immunofluorescence as indicated. Images were analysed by ImageJ with at least 100 cells counted per sample. Quantification of SEC23B overlapping with ERGIC53 was performed using the Pearson's correlation coefficient. The data are presented as mean ±SD (bottom panel). Scale bar, 10 μm. (B) U-2OS cells stably expressing FLAG-HA-tagged SEC23B were transfected with ULK1 siRNAs. Seventy-two hours after transfection, cells were fixed for immunofluorescence as indicated. Images were analysed by ImageJ with at least 100 cells counted per sample. Quantification of SEC23B overlapping with ERGIC53 was performed using the Pearson's correlation coefficient. The data are presented as mean ±SD (bottom panel). Scale bar, 10 μm. (C) HEK239T cells were transfected with a plasmid expressing Gaussia luciferase in combination with the indicated constructs. Twenty-four hours after transfection, cells were replated onto 96-well plates. After another forty-eight hours, fresh media was added to the cells, and four hours after, the culture media were collected to measure Gaussia luciferase activity. The data are presented as mean ±SD of the Gaussia luciferase activity of triplicate samples. Expression of FBXW5 was used as a positive control since it results in the downregulation of SEC23B and, therefore, it is expected to inhibit trafficking. (D) A375 cells stably expressing Gaussia luciferase were plated onto 96-well plates. After forty-eight hours, either fresh media or EBSS was added to the cells and four hours after, the culture media were collected to measure Gaussia luciferase activity. The data are presented as mean ±SD of the Gaussia luciferase activity of triplicate samples. (E) A375 parental cells or SEC23B(S186A)-A357 knock-in cells were starved with EBSS for the indicated times (±BafA1) and harvested for immunoblotting as indicated. (F) A375 parental cells or SEC23B(S186A)-A357 knock-in cells were starved with EBSS and fixed for immunofluorescence. Images of endogenous LC3 puncta underwent automated processing with at least 100 cells counted per sample. The data are presented as mean ±SD (right panel). Scale bar, 10 μm.

DOI: https://doi.org/10.7554/eLife.42253.015

The following source data and figure supplement are available for figure 5:

**Source data 1.** Source data for *Figure 5B,C,D and F*.
DOI: https://doi.org/10.7554/eLife.42253.017

**Figure supplement 1.** Ser186 in SEC23B is necessary for its localization to the ERGIC and an efficient autophagic response upon nutrient deprivation.
DOI: https://doi.org/10.7554/eLife.42253.016

The above results indicate that the presence of Ser186 in SEC23B and, presumably, its phosphorylation and consequent stabilization are required for the proper induction of autophagy in response to starvation.

## SEC24A/B, but not SEC24C/D, specifically associate with phosphorylated SEC23B and contribute to autophagy

Vertebrates express four SEC24 paralogs (*Fromme et al., 2008*), but it is not known whether any of them contribute to the regulation of the autophagic flux. We observed no difference in binding of the phospho-mimetic SEC23B(S186D) mutant with SEC24A and SEC24B (*Figure 3—figure supplement 1D*). Because the region of SEC23B that binds FBXW5 and SEC24 proteins overlaps, we investigated whether ULK1-dependent phosphorylation of SEC23B, in addition to blocking its interaction with FBXW5, affects SEC23B association with the other SEC24 paralogs. We observed that whereas both wild-type SEC23B and SEC23B(S186A) associated with all four SEC24 paralogs, SEC23B (S186D) only interacted with SEC24A and SEC24B (*Figure 6A–B*). Similarly, SEC24B, but not SEC24C, interacted with endogenous phosphorylated SEC23B (*Figure 6C*). Accordingly, molecular dynamics simulations showed that, in contrast to what observed with SEC23B(S186D) and SEC24A, there was a significant loss in the average number of contacts between SEC23B(S186D) and SEC24C, compared to contacts observed with wild-type SEC23B (*Figure 6D*). We also performed Molecular Mechanics/Generalized Born Surface Area calculations, which predicted a significantly higher value of binding free energy for the SEC23B(S186D)-SEC24C mutant dimer, resulting in the following trend: SEC23B-SEC24A ~ SEC23B-SEC24C $\lesssim$ SEC23B(S186D)-SEC24A << SEC23B (S186D)-SEC24C (*Figure 6D*). These observations indicate a lack of stability of the SEC23B(S186D)-SEC24C system compared to the other three, in agreement with our experimental results.

Altogether, these results suggest that upon serum starvation, the population of phosphorylated SEC23B preferentially associates with SEC24A and SEC24B.

Notably, co-depletion of both SEC24A and SEC24B, but not SEC24C and SEC24D, inhibited the increase in autophagic flux induced by the silencing of FBXW5 (*Figure 6E* and *Figure 4—figure supplement 1B*), similar to what we observed by co-depleting SEC23B together with FBXW5 (*Figure 4C*). We also evaluated by immunofluorescence the colocalization of SEC24 family members with ERGIC53. Upon starvation, SEC24B, but not SEC24C, colocalized more with ERGIC53 and this event was not observed in SEC23B(S186A) knock-in cells (*Figure 6F*), suggesting that the localization of SEC24B to the ERGIC is regulated by its association with phospho-SEC23B.

Thus, our data demonstrate the existence of a high degree of specificity among SEC24 paralogs with respect to their ability to contribute to the autophagic flux.

## A human melanoma-associated mutation in SEC23B results in its stabilization and increased autophagy flux

By searching for mutations of SEC23B in publicly available human cancer databases, we found a mutation in human melanoma that converts S186 of SEC23B to asparagine (ID#: COSM5391641). We thus generated SEC23B(S186N), a mutant that mimics the cancer associated mutation, and observed a reduced binding to FBXW5, SEC24C, and SEC24D (*Figure 7A*) but not to SEC24A and SEC24B, similar to what observed with SEC23B(S186D) (*Figure 6A*). Moreover, SEC23B(S186N) displayed a longer half-life than wild-type SEC23B (*Figure 6B*). These results suggest that SEC23B (S186N), by being stable, may promote autophagy. To test this hypothesis, we used CRISPR/Cas9 to generate SEC23B(S186N) A375 melanoma cells (*Figure 5—figure supplement 1B*). Automated quantification of endogenous LC3 showed that, compared to parental cells, the SEC23B(S186N) knock-in cells displayed more autophagy during unstressed conditions (*Figure 7C*). The number of LC3 puncta were also increased in the presence of bafilomycin A1 (*Figure 7C*), indicating that SEC23B(S186N) increases the autophagic flux, and that the constitutive increase in LC3 puncta is not due to inhibition of the lysosome-dependent degradation of autophagosomes.

These results show that the SEC23B(S186N) mutation found in melanoma mimics SEC23B phosphorylated on Ser186 similarly to what observed for SEC23B(S186D). In fact, SEC23B(S186N) does not bind efficiently FBXW5, SEC24C, and SEC24D, and promotes autophagy, in agreement with a possible advantageous role resulting from increasing autophagic flux to ensure tumour cell homeostasis (*Rybstein et al., 2018*).

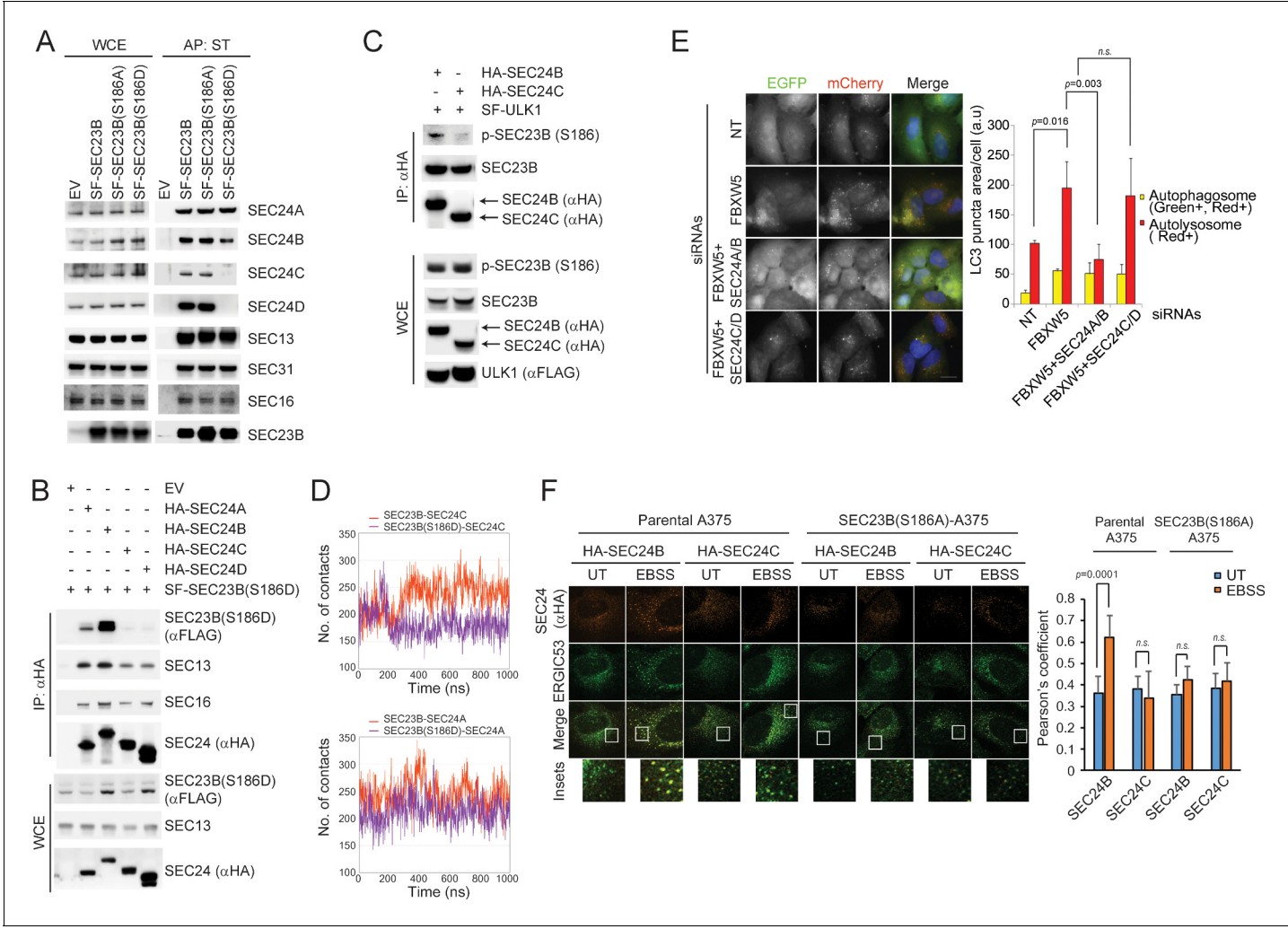

**Figure 6.** SEC24A/B, but not SEC24C/D, specifically associate with phosphorylated SEC23B and contribute to autophagy. (**A**) HEK293T cells were transfected with an EV, SF-SEC23B, or SF-SEC23B mutants. Twenty-four hours after transfection, cells were treated with MLN4924 for 4 hr before harvesting them for affinity-purification (AP) with Streptactin (ST) beads and immunoblotting. (**B**) HEK293T cells were transfected with an EV or SF) SEC23B(S186D) in combination with either HA-tagged SEC24 paralogs as indicated. Twenty-four hours after transfection, cells were treated with MLN4924 for 4 hr before harvesting them for immunoprecipitation (IP) with anti-HA beads and immunoblotting. (**C**) HEK293T cells were transfected with either HA-tagged SEC24B or HA-tagged SEC24C together with SF-ULK1. Twenty-four hours after transfection, cells were treated with MLN4924 for 4 hr before harvesting them for immunoprecipitation (IP) with anti-HA beads and immunoblotting. (**D**) Evolution of the inter-molecular contacts between monomers in the four studied systems during 1 μs molecular dynamics simulations. Upper panel: SEC23B-SEC24C (red), SEC23B(S186D)-SEC24C (violet); bottom panel: SEC23B-SEC24A (red), SEC23B(S186D)-SEC24A (violet). Contacts were calculated as the number of heavy atom interacting pairs within a distance of 4.4 Å. (**E**) U-2OS cells stably expressing tandem fluorescent-tagged LC3 (pBabe-mCherry-EGFP-LC3) were transfected with a NT oligo or a FBXW5 siRNA oligo in combination with the indicated siRNA oligos. Forty-eight hours after transfection, cells were replated onto coverglass followed by fixation twenty-four hours after replating. Images of mCherry-EGFP-LC3 puncta underwent automated processing with at least 100 cells counted per sample. The data are presented as mean ±SD (right panel). Scale bar, 10 μm (**F**) A375 parental cells or SEC23B(S186D)-A357 knock-in cells were transfected with either HA-tagged SEC24B or HA-tagged SEC24C. Twenty-four hours after transfection, cells were either left untreated (UT) or starved with EBSS for two hours. Next, cells were fixed for immunofluorescence as indicated. Images were analysed by ImageJ with at least 100 cells counted per sample. Quantification of SEC24 overlapped with ERGIC53 was performed using the Pearson's correlation coefficient. The data are presented as mean ±SD (right panel). Scale bar, 10 μM.

DOI: https://doi.org/10.7554/eLife.42253.018

The following source data is available for figure 6:

**Source data 1.** Source data for *Figure 6E and F*.
DOI: https://doi.org/10.7554/eLife.42253.019

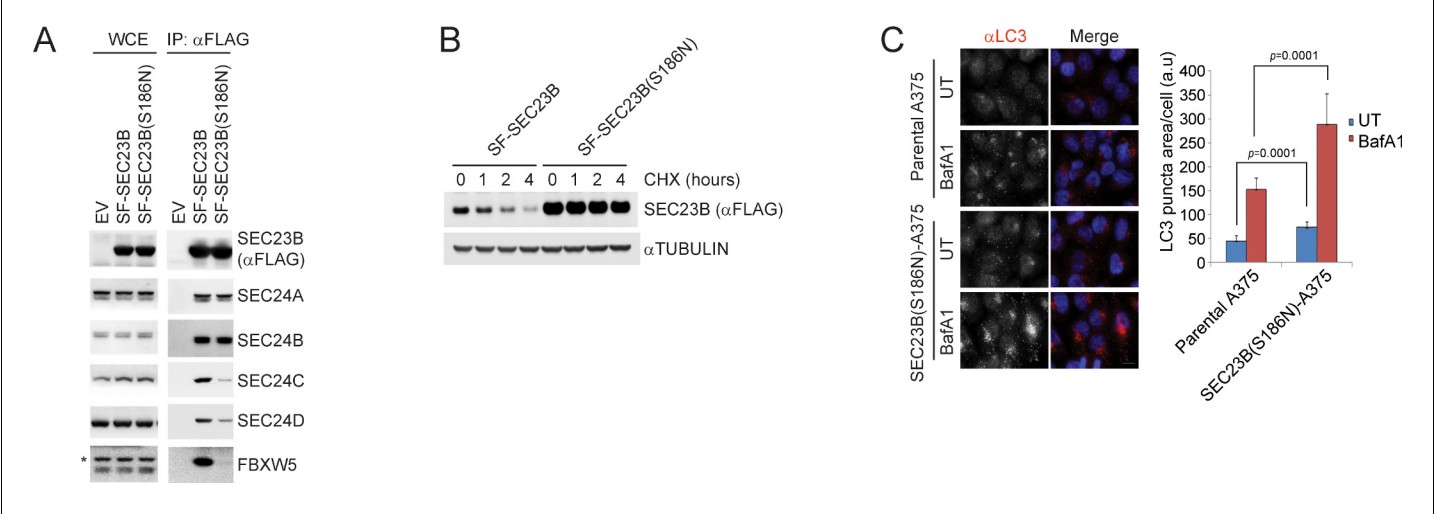

**Figure 7.** A human melanoma-associated mutation in SEC23B results in its stabilization and increased autophagy flux. (**A**) HEK239T cells were transfected with either SF-SEC23B or SF-SEC23B(S186N). Twenty-four hours after transfection, cells were harvested for immunoprecipitation (IP) with FLAG-M2 beads and immunoblotted as indicated. (**B**) HEK293T cells were transfected with either Streptag-FLAG-tagged (SF) wild-type SEC23B (SF-SEC23B) or Streptag-FLAG-tagged (SF) SEC23B(S186N). Twenty-four hours after transfection, cells were treated with cycloheximide (CHX) for the indicated time and subjected to immunoblot analysis. (**C**) A375 parental cells or SEC23B(S186N)-A357 knock-in cells were treated with BafA1 and fixed for immunofluorescence. Images of endogenous LC3 puncta underwent automated processing with at least 100 cells counted per sample. The data are presented as mean ±SD (right panel). Scale bar, 10 μm.

DOI: https://doi.org/10.7554/eLife.42253.020

The following source data is available for figure 7:

**Source data 1.** Source data for *Figure 7C*.
DOI: https://doi.org/10.7554/eLife.42253.021

## Discussion

SEC23B is an essential component of the COPII multi-subunit protein complex that is responsible for the transport of cargo proteins destined to be secreted (*Fromme et al., 2008*; *Zanetti et al., 2011*). Although COPII vesicles were originally thought to participate exclusively in the secretory pathway, growing evidence demonstrates that they also play an important role in controlling and executing the autophagic cascade (see Introduction). However, the molecular mechanisms that repurpose COPII for the autophagic process have remained largely unknown. In this study, we demonstrated that FBXW5 targets SEC23B for degradation to limit autophagy during basal, unperturbed conditions. Upon induction of autophagy by nutrient deprivation, ULK1 phosphorylates Ser186 in SEC23B, inhibiting its interaction with and degradation via FBXW5. The resulting increased levels of SEC23B are crucial for proper autophagic flux. We also found that phosphorylation of Ser186 in SEC23B inhibits its binding to SEC24C and SEC24D, but not SEC24A and SEC24B. Accordingly, depletion of SEC24A and SEC24B, but not SEC24C and SEC24D rescue the phenotype (i.e., the increase in autophagy during non-stressed conditions) induced by FBXW5 silencing. The fact that the phosphorylation of Ser186 inhibits the binding of SEC23B to SEC24C and SEC24D may explain, at least in part, the reduced secretion observed upon nutrient starvation. It would be interesting to understand whether Ser186 phosphorylation also modulates the GAP activity of SEC23B towards SAR1.

It has been suggested that autophagic COPII vesicles can be distinguished from trafficking COPII vesicles by their different site of generation, that is autophagic COPII vesicles are generated from the ERGIC rather than the ERES, and they are potent stimulators of LC3 lipidation in response to starvation (*Ge et al., 2015*; *Ge et al., 2014*). PI3KC3, which is activated by ULK1, is required for the relocation of COPII components to the ERGIC (*Ge et al., 2015*; *Ge et al., 2014*). However, the differential molecular composition between trafficking COPII vesicles and autophagic COPII vesicles remain largely unknown. Our results indicate that at least a subpopulation of autophagic COPII vesicles present at the ERGIC upon starvation contain SEC23B phosphorylated on Ser186 by ULK1.

Moreover, we show that SEC24B, but not SEC24C, colocalizes more abundantly with the ERGIC, and this event depends on the presence of Ser186 in SEC23B. Interestingly, in yeast, phosphorylation of Sec24 by casein kinase one promotes its binding to autophagy factors to increase autophagosome abundance (*Davis et al., 2016*), which is reminiscent of what we found in human cells (i.e., phosphorylation of SEC23B by ULK1 induces the localization of SEC23B and, consequently, SEC24A and SEC24B to the ERGIC to promote the autophagic flux).

Recently, it has been shown that, in presence of nutrients, the basal activity of ULK1 mediates the phosphorylation of SEC16A (promoting the assembly of COPII complexes at the ERES) and that, in response to starvation, the activation of ULK1 induces the dissociation of SEC23A from SEC31A (inhibiting the secretory pathway) (*Joo et al., 2016*; *Gan et al., 2017*). Our study shows that, in the absence of nutrients, activated ULK1 promotes the phosphorylation of SEC23B on Ser186, inducing SEC23B accumulation and the formation of autophagic COPII vesicles. Moreover, the decrease in the abundance of the SEC23B-SEC24C/D complexes likely contributes to inhibit the secretory pathway.

Altogether, our results provide evidence for one of the molecular mechanisms by which ULK1 functions as a switch necessary to commit COPII to autophagy in response to starvation.

# Materials and methods

## Key resources table

| Reagent type (species) or resource | Designation | Source or reference | Identifiers | Additional information |
|---|---|---|---|---|
| Antibody | FBXW5 | This paper | Yenzyme | (1:250 dilution for western blotting) |
| Antibody | Phospho-SEC23B (S186) | This paper | Yenzyme | (1:200 dilution for western blotting) |
| Antibody | SEC23A | Randy Schekman's lab | PMID: 24069399 | (1:1000 dilution for western blotting) |
| Antibody | SEC23B | Randy Schekman's lab | PMID: 24069399 | (1:1000 dilution for western blotting) |
| Antibody | SEC24A | Randy Schekman's lab | PMID: 18056412 | (1:2000 dilution for western blotting) |
| Antibody | SEC24B | Randy Schekman's lab | PMID: 18056412 | (1:2000 dilution for western blotting) |
| Antibody | SEC24C | Randy Schekman's lab | PMID: 18056412 | (1:2000 dilution for western blotting) |
| Antibody | SEC24D | Randy Schekman's lab | PMID: 18056412 | (1:2000 dilution for western blotting) |
| Antibody | FLAG | Sigma-Aldrich | Cat# F7425 | (1:1000 dilution for western blotting, 1:1000 for immunofluorescence) |
| Antibody | HA | Covance | Cat# MMS-101P | (1:1000 dilution for western blotting) |
| Antibody | MYC | Sigma-Aldrich | Cat# M5546 | (1:1000 dilution for western blotting) |
| Antibody | SAR1 | Thermo Scientific | Cat# PA5-27642 | (1:5000 dilution for western blotting) |
| Antibody | CUL1 | Life Technologies | Cat# 322400 | (1:1000 dilution for western blotting) |
| Antibody | SKP1 | Michele Pagano's lab | Yenzyme | (1:1000 dilution for western blotting) |
| Antibody | SEC13 | Bethyl Laboratories | Cat# A303-980A | (1:10,000 dilution for western blotting) |

*Continued on next page*

*Continued*

| Reagent type (species) or resource | Designation | Source or reference | Identifiers | Additional information |
|---|---|---|---|---|
| Antibody | SEC31A | Bethyl Laboratories | Cat# A302-336A | (1:10,000 dilution for western blotting) |
| Antibody | ULK1 | Cell Signaling Technology | Cat# 8054 s | (1:1000 dilution for western blotting) |
| Antibody | Phospho-ULK1 (S757) | Cell Signaling Technology | Cat# 6888 s | (1:1000 dilution for western blotting) |
| Antibody | LC3B | Novus Biologicals | Cat# NB100-2220 | (1:10,000 dilution for western blotting, 1:1000 for immuno fluorescence) |
| Antibody | Phospho-ATG13 (S318) | Rockland Immuno chemicals | Cat# 600–401 C49S | (1:1000 dilution for western blotting) |
| Antibody | Phospho-Beclin-1 (S15) | Abbiotec | Cat# 254515 | (1:500 dilution for western blotting) |
| Antibody | Beclin-1 | Santa Cruz Biotechnology | Cat# SC-48381 | (1:1000 dilution for western blotting) |
| Antibody | GAPDH | Cell Signaling Technology | Cat# 97166S | (1:10,000 dilution for western blotting) |
| Antibody | ACTIN | Sigma-Aldrich | Cat# A5441 | (1:10,000 dilution for western blotting) |
| Antibody | Tubulin | Sigma-Aldrich | Cat# T6074-200UL | (1:10,000 dilution for western blotting) |
| Antibody | ERGIC53 | Sigma-Aldrich | Cat# E1031-200UL | (1:5000 dilution for western blotting, 1:1000 for immuno fluorescence) |
| Antibody | SEC16 | Bethyl Laboratories | Cat# A300-648A | (1:5000 dilution for western blotting) |
| Antibody | FBXO11 | Novus Biologicals | Cat# H00080204-B01 | (1:1000 dilution for western blotting) |
| Antibody | FBXW9 | Michele Pagano's lab | | (1:1000 dilution for western blotting) |
| Peptide, recombinant protein | MG132 | Peptide international | Cat# IZL-3175v | (final concentration, 5 mM) |
| Chemical compound, drug | MLN4924 | Active Biochem | Cat# A-1139 | (final concentration, 1 mM) |
| Chemical compound, drug | Cycloheximide | Sigma-Aldrich | Cat# C7698-1G | (final concentration, 50 ng/ml) |
| Chemical compound, drug | Bafilomycin A1 | Santa Cruz Biotechnology | Cat# sc-201550A | (final concentration, 0.1 mg/ml) |
| Chemical compound, drug | Polybrene | Sigma-Aldrich | Cat# H9268-10G | (final concentration, 8 µg/ml) |
| Chemical compound, drug | SBI-0206965 | Selleck Chemicals | Cat# S7885 | (final concentration, 1 mM) |
| Peptide, recombinant protein | SEC23B peptide | This paper | Yenzyme | |

*Continued on next page*

*Continued*

| Reagent type (species) or resource | Designation | Source or reference | Identifiers | Additional information |
|---|---|---|---|---|
| Peptide, recombinant protein | SEC23B phospho-peptide | This paper | Yenzyme | |
| Cell line (human) | HEK293T | ATCC | Cat# CRL-3216 | |
| Cell line (human) | U-2OS | ATCC | Cat# HTB-96 | |
| Cell line (human) | RPE1-hTERT | ATCC | Cat# CRL-4000 | |
| Cell line (human) | A375 | ATCC | Cat# CRL-1619 | |
| Cell line (human) | HeLa | ATCC | Cat# CCL-2 | |
| Commercial assay or kit | Pierce Gaussia Luciferase Flash Assay Kit | Thermo Scientific | Cat# 16159 | |
| Other | siRNAs to FBXW5 (#1) | Dharmacon | Cat# J-013389 -05-0002 | Oligonucleotides |
| Other | siRNAs to FBXW5 (#2) | Dharmacon | Cat# J-013389 -06-0002 | Oligonucleotides |
| Other | siRNAs to FBXW5 (#3) | Dharmacon | Cat# J-013389 -07-0002 | Oligonucleotides |
| Other | siRNAs to FBXW5 (#4) | Dharmacon | Cat# J-013389 -08-0002 | Oligonucleotides |
| Other | siRNAs to FBXW5 | Santa Cruz Biotechnology | Cat# sc-92629 | Oligonucleotides |
| Other | siRNAs to SEC23A | Dharmacon | Cat# M-009582 -00-0005 | Oligonucleotides |
| Other | siRNAs to SEC23B | Dharmacon | Cat# M-009592 -01-0005 | Oligonucleotides |
| Other | siRNAs to ULK1 | Santa Cruz Biotechnology | Cat# sc-44182 | Oligonucleotides |
| Other | siRNAs to ULK2 | Santa Cruz Biotechnology | Cat# sc-44183 | Oligonucleotides |
| Other | siRNAs to SEC24A | Dharmacon | Cat# L-024405 -01-0005 | Oligonucleotides |
| Other | siRNAs to SEC24B | Dharmacon | Cat# L-008299 -02-0005 | Oligonucleotides |
| Other | siRNAs to SEC24C | Dharmacon | Cat# L-008467 -02-0005 | Oligonucleotides |
| Other | siRNAs to SEC24D | Dharmacon | Cat# L-008493 -01-0005 | Oligonucleotides |
| Other | Non-targeting siRNA (CGUACGCGGA AUACUUCGA) | Dharmacon | | Oligonucleotides |
| Recombinant DNA reagent | pCS2 + 3x HA-Sec24A | Randy Schekman's lab | | |
| Recombinant DNA reagent | pCS2 + 3x HA-Sec24B | Randy Schekman's lab | | |
| Recombinant DNA reagent | pCS2 + 3x HA-Sec24C | Randy Schekman's lab | | |
| Recombinant DNA reagent | pCS2 + 3x HA-Sec24D | Randy Schekman's lab | | |
| Recombinant DNA reagent | pBabe-puro-mCherry-EGFP -LC3B | Addgene | Cat# 22418 | |

*Continued on next page*

*Continued*

| Reagent type (species) or resource | Designation | Source or reference | Identifiers | Additional information |
|---|---|---|---|---|
| Recombinant DNA reagent | pcdna6.2- myc ULK1 wt | Addgene | Cat# 27629 | |
| Recombinant DNA reagent | pcdna6.2- myc ULK1 k46I | Addgene | Cat# 27630 | |
| Recombinant DNA reagent | GFP-SAR1 (T39N) | Antonella De Matteis' lab | | |
| Recombinant DNA reagent | GFP-SAR1 (H79G) | Antonella De Matteis' lab | | |

## Cell lines and drug treatments

Cell lines were purchased from ATCC and were routinely checked for mycoplasma contamination with the Universal Mycoplasma Detection Kit (ATCC 30–1012K). All cells were maintained in DMEM/GlutaMAX supplemented with 10% fetal bovine serum (FBS) and penicillin/streptomycin. For nutrient starvation, DMEM and FBS were removed and cells were grown in EBSS (Sigma) for the indicated times.

## Biochemical methods

For immunoprecipitation, cell extracts were prepared using lysis buffer (50 mM Tris pH 7.4, 150 mM NaCl, 2 mM EDTA, 10% glycerol, 0.5% NP-40, protease inhibitors, and phosphatase inhibitors), followed by incubation with Streptactin beads (IBA) or FLAG-M2 beads (Sigma) for 2 hr at 4°C. For immunoblotting, each sample was solubilized with lysis buffer (50 mM Tris pH 7.4, 150 mM NaCl, 2 mM EDTA, 10% glycerol, 0.5% NP-40, protease inhibitors, and phosphatase inhibitors). Cell extracts were quantified using BCA protein assay kit (Pierce) and solubilized with LDS-sample buffer (Life technology) followed by boiling at 95°C for five minutes.

## Immunofluorescence microscopy

Immunofluorescence microscopy was performed as described previously (*Jeong et al., 2013*). Briefly, cells were cultured on round coverglass in 24-well culture dishes. After the indicated treatments, cells were washed with PBS followed by fixation with either cold methanol or 4% PFA/PBS. Cells were then permeabilized for 15 min with 3% BSA in 0.5% Triton X-100/PBS. Primary antibodies were incubated for one hour at room temperature, and secondary antibodies conjugated to either Alexa Fluor 488 or Alexa Fluor 555 were incubated for one hour at room temperature in 3% BSA/0.1% Triton X-100/PBS. Coverglasses were mounted on slideglass using Pro-long Gold anti-fading reagent with DAPI (Molecular probes).

## Affinity purification and mass spectrometry

Affinity purification and mass spectrometry were performed as described previously (*Jeong et al., 2013*). Briefly, Streptag-FLAG-tagged FBXW5 was transiently transfected into HEK293T cells. Cells were treated with MLN4924 for four hours prior to harvest, and then solubilized with lysis buffer (50 mM Tris pH 7.4, 150 mM NaCl, 2 mM EDTA, 10% glycerol, 0.5% NP-40, protease inhibitors, and phosphatase inhibitors). Cell extracts were immunoprecipitated with either Streptactin beads (IBA) or FLAG-M2 beads (Sigma). Immunoprecipitation and subsequent mass spectrometry was carried out as previously described (*Kuchay et al., 2017*).

## Plasmids, siRNA and shRNA

FBXW5 and SEC23B mutants were generated using KAPA HiFi polymerase (Kapabiosystems). All cDNAs were subsequently sequenced. ULK1 and ULK1(K46I) plasmids were purchased from Addgene. Sar1 plasmids were generously provided by Dr. Antonella De Matteis. SEC24A, B, C, and D plasmids were generously provided by Dr. Randy Schekman. ON-Target siRNAs targeting FBXW5, SEC23B, SEC24A, SEC24B, SEC24C, SEC24D were purchased from Dharmacon. The production of lentivirus was previously described (*Jeong et al., 2013*). ULK1 and ULK2 siRNA oligos and pooled

FBXW5 siRNA oligos were purchased from Santa Cruz Biotechnology. Non-targeting siRNA oligo (CGUACGCGGAAUACUUCGA) served as a negative control.

## Antibodies

An anti-FBXW5 antibody was generated by immunizing rabbits with FBXW5 peptides (Yenzyme) and affinity-purified using the same peptides immobilized on CNBr-sepharose. A rabbit polyclonal antibody against phospho-S186-SEC23B was generated and affinity purified by YenZym Antibodies. Anti- SEC23B, SEC24A, SEC24B, SEC24C, and SEC24D antibodies were provided by Dr. Randy Schekman. Mouse monoclonal antibodies were from Sigma-Aldrich (anti-FLAG M2), Covance (anti-HA), and Thermo Scientific (Sar1). Rabbit polyclonal antibodies were from Invitrogen (CUL1 and SKP1), Bethyl Laboratories, Inc. (Sec13 and SEC31A), Cell Signaling Technology (ULK1 and phospho-ULK1 (S757)), Novus Biological (LC3B), Rockland (phospho-ATG13 (S318)), and Abbiotec (phospho-Beclin-1 (S15).

## In vitro kinase assay

Kinase and substrates were purified from HEK293T cells that had been transfected with plasmids expressing individual kinase and substrates. Kinase reaction buffer (KRB; 20 mM Tris, pH 7.5, 20 mM MgCl$_2$, 25 mM β-glycerophosphate, 2 mM dithiothreitol and 100 μM sodium orthovanadate) were used to elute the purified proteins. Kinase and substrates were mixed and incubated at a final volume of 20 μL in KRB containing 20 μM ATP, 5 μg substrates at 30°C for 60 min. The reaction was stopped by the addition of sample buffer, boiled and analysed by immunoblot with phospho-SEC23B (Ser186) specific antibody.

## Computational methods

The SEC24A and SEC24C monomers were extracted from the Protein Data Bank (PDB) (entries 3EGD and 3EH2, respectively), while SEC23B was modeled by homology from the PDB structure of SEC23A (entry 5KYN), including the missing residues from the crystal structure. The structure of the mutant SEC23B(S186D) was obtained using PyMol Mutagenesis Wizard tool (Edn. Version 1.7. Schrodinger, LLC, 2013). MD simulations were performed with the GROMACS v5.1 package (Abraham et al., 2015) using the Amber99SB force field (Hornak et al., 2006); the starting conformation for each dimer was modeled from the structure of the SEC23A-SEC24A dimer (PDB 3EGD). Only residues 120 to 405 from SEC23B, 502 to 742 from SEC24, and 173 to 419 from SEC24C were used in the simulations, as these are the domains involved in the dimer interaction, and we have seen that their structures were not influenced by the rest of the protein (data not shown). The systems were solvated with the SPCE water model in a triclinic box, extending at least 10 Å from every atom of the protein, and neutralized adding sufficient Na and Cl counter ions to reach 0.15 M concentration. Bond lengths were constrained using the LINCS algorithm allowing a 2fs time-step. Long-range electrostatics interactions were taken into account using the particle-mesh Ewald (PME) approach. The non-bonded cut-off for Coulomb and Van der Waals interactions were both 10 Å, and the non-bonded pair list was updated every 25 fs. Energy minimization was conducted through the steepest-descent algorithm, until the maximum force decayed to 1000 [kJ mol$^{-1}$ nm$^{-1}$]. The equilibration stage of the whole system consisted in 500 ps of NVT simulation followed by 500 ps of NPT simulation. Temperature was kept constant at 310 K using a modified Berendsen thermostat (Essmann et al., 1995) with a coupling constant of 0.1 ps. Constant pressure of 1 bar was applied in all directions with a coupling constant of 2.0 ps and a compressibility of 4.5 10$^{-5}$ bar$^{-1}$. Finally, the equilibrated systems were subjected to a 1 μs MD simulation run each at 310 K. Binding free energies were calculated using de MM-GBSA scheme (Genheden and Ryde, 2015) provided in the Amber16 package (Case et al., 2017) using the single trajectory approach. Hundred snapshots were collected at time intervals of 5 ns from the last 500 ns of the MD simulations, thus guaranteeing statistical independence (Anisimov and Cavasotto, 2011). The salt concentration was set to 0.150 M, and the dielectric constants to 80 and 1 for the solvent and the proteins, respectively.

## CRISPR genome editing

To generate SEC23B S186A and S186N knock-in cells, an optimal gRNA target sequence closest to the genomic target site and a 2.1 kb homologous recombination (HR) donor template were

designed using the Benchling CRISPR Genome Engineering tool. The HR donor template was designed to introduce alanine or asparagine substitutions at position S186, and a silent mutation to introduce a KpnI restriction site for genotyping. SEC23B gRNA target sequence (see; *Figure 5—figure supplement 1B*) was cloned into pSpCas9(BB)−2A-GFP (PX458), a gift from F. Zhang (Addgene plasmid no. 48138) (*Ge et al., 2013*). A375 cells were seeded into 10 cm dishes at approximately 70% confluency and transfected with 2.5 μg each of gRNA-containing PX458 plasmid and HR donor template, using lipofectamine 3000 (Life Technologies). The transfection was performed according to the manufacturer's recommended protocol, using a 2:1 ratio of lipofectamine:DNA. Two days after transfection, GFP-positive cells were sorted using the Beckman Coulter MoFlo XDP cell sorter (100 μm nozzle), and 5,000 cells were plated on a 15 cm dish. About a week later, single-cell clones were picked, trypsinized in 0.25% Trypsin-EDTA for 5 min, and plated into individual wells of a 96-well plate for genotyping. Genomic DNA was collected using QuickExtract (Epicentre). Genotyping PCRs were performed with MyTaq HS Red Mix (Bioline), using primers surrounding the genomic target sites. The resulting PCR products were then sequenced and aligned to the corresponding wild-type template in Benchling to determine the presence of a recombination event.

## Gaussia luciferase assay

To measure the activity of the secretory pathway in HEK293T cells transfected with various cDNAs, a pCMV-*Gaussia* luciferase plasmid (ThermoFisher Scientific) was co-transfected to be used as a reporter by measuring luciferase activity in the conditioned medium (*Badr et al., 2007*). The activity of the secreted Gaussia luciferase was measured using Pierce Gaussia luciferase Flash Assay Kit (ThermoFisher Scientific) according to the manufacturer's instruction.

## Membrane fractionation

Membrane fractionation was performed through a modified protocol based on *Ge et al.* (*Ge et al., 2013*). HeLa cells (ten 15 cm dishes) were cultured to 95% confluence. Cells were treated with 20 μg/ml digitonin (5 min on ice) in B88 (20 mM Hepes, pH 7.2, 250 mM sorbitol, 150 mM potassium acetate, and 5 mM magnesium acetate). Membranes were pelleted at 300xg, washed in B88, and incubated with 3 mM GMPPNP and purified human COPII proteins [10 μg SAR1B and 10 μg of either SEC23B or SEC23B(S186D)], which were purified as described (*Kim et al., 2005*). Membranes were incubated for 30 min at 37°C and subjected to sequential differential centrifugation at 1,000 × $g$ (10 min), 3,000 × $g$ (10 min), 25,000 × $g$ (20 min) to collect the membranes sedimented at each speed. The 25,000 × $g$ membrane pellet was suspended in 0.75 ml 1.25 M sucrose buffer and overlayed with 0.5 ml 1.1 M and 0.5 ml 0.25 M sucrose buffer (Golgi isolation kit; Sigma). Centrifugation was performed at 120,000 × $g$ for 2 hr (TLS 55 rotor, Beckman), after which the interface between 0.25 M and 1.1 M sucrose (L fraction) was selected and suspended in 1 ml 19% OptiPrep for a step gradient containing 0.5 ml 22.5%, 1 ml 19% (sample), 0.9 ml 16%, 0.9 ml 12%, 1 ml 8%, 0.5 ml 5% and 0.2 ml 0% OptiPrep each. Each density of OptiPrep was prepared by diluting 50% OptiPrep (20 mM Tricine-KOH, pH 7.4, 42 mM sucrose and 1 mM EDTA) with a buffer containing 20 mM Tricine-KOH, pH 7.4, 250 mM sucrose and 1 mM EDTA. The OptiPrep gradient was centrifuged at 150,000 × $g$ for 3 hr (SW 55 Ti rotor, Beckman) and subsequently fractions of 0.5 ml each, were collected from the top. The fractions were then analyzed by immunoblot.

## Quantification and statistical analysis

Images were analyzed with an in-house developed python script. Cell counts were calculated by detecting nuclei labelled by DAPI in the blue channel of the image. First, a Gaussian smoothing was applied, and then an Otsu threshold to determine the nuclei mask. This was followed by watershed segmentation to separate touching nuclei, as well as filtering based on area to remove small spots. Partial nuclei touching image borders were included but were counted as a fraction based on the average nuclei size measured in the data set. The puncta (green channel) were detected using the Laplacian of Gaussian (LoG) blob detection algorithm as provided by the python package 'scikit-image' (*van der Walt et al., 2014*). Any blobs found in the regions covered by cell nuclei were ignored. The puncta area was calculated by finding the area of a circle with radius proportional to the standard deviation of the Gaussian kernel that detected the blob, as returned by the LoG algorithm. Prior to applying LoG, some images required a median filter for removal of speckle noise.

Some very bright cells were removed from the analysis (identified nuclei and a surrounding area), since it was not possible to distinguish puncta in these areas. These cells were identified by finding outliers from the set of nuclei mean intensities of the image. All statistical analysis was performed with unpaired Student's t test, and it is considered significant when the p value is less than 0.05. n.s., not significant. Data were expressed as mean ±SD of at least three independent experiments performed in triplicate.

## Acknowledgements

The authors thank A D Matteis and R Schekman for reagents; MP and YTJ are grateful to TM Thor and SO Hong, respectively, for continuous support. This work was funded by grants from the National Institute of Health (R01-CA076584 and R01-GM057587) to MP, and Agencia Nacional de Promoción Científica y Tecnológica-Argentina (PICT-2014–0458, PICT2016-2620) to MR, and (PICT-2014–3599) to CNC CNC thanks the National System of High-Performance Computing (Sistemas Nacionales de Computación de Alto Rendimiento, SNCAD) and the Computational Centre of High-Performance Computing (Centro de Computación de Alto Rendimiento, CeCAR) for granting use of their computational resources. MP is an Investigator with the Howard Hughes Medical Institute.

## Additional information

### Funding

| Funder | Grant reference number | Author |
| --- | --- | --- |
| National Institutes of Health | R01-CA076584 | Michele Pagano |
| National Institutes of Health | R01-GM057587 | Michele Pagano |
| Agencia Nacional de Promoción Científica y Tecnológica | PICT-2014-0458 | Mario Rossi |
| Agencia Nacional de Promoción Científica y Tecnológica | PICT2016-2620 | Mario Rossi |
| Agencia Nacional de Promoción Científica y Tecnológica | PICT- 2014-3599 | Claudio N Cavasotto |

The funders had no role in study design, data collection and interpretation, or the decision to submit the work for publication.

### Author contributions

Yeon-Tae Jeong, Conceptualization, Resources, Data curation, Software, Formal analysis, Validation, Investigation, Visualization, Methodology, Writing—original draft, Writing—review and editing; Daniele Simoneschi, Resources, Investigation, Visualization, Writing—review and editing; Sarah Keegan, Software, Formal analysis, Investigation; David Melville, Resources, Formal analysis, Validation, Investigation, Visualization; Natalia S Adler, Formal analysis, Investigation, Visualization; Anita Saraf, Data curation, Formal analysis; Laurence Florens, Michael P Washburn, Supervision, Writing—review and editing; Claudio N Cavasotto, David Fenyö, Software, Supervision, Writing—review and editing; Ana Maria Cuervo, Resources, Supervision, Writing—review and editing; Mario Rossi, Supervision, Writing—original draft, Writing—review and editing; Michele Pagano, Conceptualization, Resources, Supervision, Funding acquisition, Writing—original draft, Project administration, Writing—review and editing

### Author ORCIDs

Yeon-Tae Jeong ◍ http://orcid.org/0000-0002-1104-1161
Michael P Washburn ◍ http://orcid.org/0000-0001-7568-2585
Claudio N Cavasotto ◍ https://orcid.org/0000-0002-1372-0379
David Fenyö ◍ http://orcid.org/0000-0001-5049-3825
Michele Pagano ◍ http://orcid.org/0000-0003-3210-2442

Decision letter and Author response
Decision letter https://doi.org/10.7554/eLife.42253.026
Author response https://doi.org/10.7554/eLife.42253.027

## Additional files

### Supplementary files

• Transparent reporting form
DOI: https://doi.org/10.7554/eLife.42253.022

### Data availability

All data generated or analysed during this study are included in the manuscript and supporting files. Mass spectrometry data is available at http://www.stowers.org/research/publications/libpb-1118 (ftp://odr.stowers.org/LIBPB-1118) and has also been deposited to the MassIVE repository. Source data files have been provided for Figures 1, 3, 4, 5, 6, 7, Figure 2—figure supplement 1, and Figure 4—figure supplement 1.

The following dataset was generated:

| Author(s) | Year | Dataset title | Dataset URL | Database and Identifier |
|---|---|---|---|---|
| Jeong Y-T, Simoneschi D, Keegan S, Melville D, Adler NS, Saraf A, Florens L, Washburn MP, Cavasotto CN, Fenyö D, Cuervo A-M, Rossi M, Pagano M | 2018 | MudPIT analyses of the proteins associated with FBXW5 in HEK293T cells | http://proteomecentral.proteomexchange.org/cgi/GetDataset?ID=PXD012197 | MassIVE, PXD012197 |

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
