## [Decision Letter]

Thank you for submitting your article "The ULK1-FBXW5-SEC23B nexus controls autophagy" for consideration by *eLife*. Your article has been reviewed by three peer reviewers, one of whom is a member of our Board of Reviewing Editors, and the evaluation has been overseen by Ivan Dikic as the Senior Editor. The following individuals involved in review of your submission have agreed to reveal their identity: Catherine Rabouille (Reviewer #3).

The reviewers have discussed the reviews with one another and the Reviewing Editor has drafted this decision to help you prepare a revised submission.

Summary:

The authors identified SEC23B as a substrate of the E3 ligase FBXW5. They found that FBXW5 leads to proteasomal degradation of SEC23B under normal conditions. In contrast, under starvation conditions, ULK1 phosphorylates SEC23B on Ser186, preventing recognition by FBXW5 and redirecting SEC23B to the ERGIC together with SEC24A/B. The data are clearly demonstrated, and the logic is easy to follow.

Essential revisions:

1) The authors propose that ULK1-mediated phosphorylation of SEC23B promotes its translocation to the ERGIC, but this is not directly shown. The authors should test whether the translocation of SEC23B to the ERGIC is indeed abolished in ULK1- or ULK1/2-knockdown cells.

2) The authors conclude that FBXW5-dependent degradation of SEC23B downregulates basal autophagy based on the results of the LC3 puncta formation and mCherry-GFP-LC3 assays. However, the authors should rule out the possibility that SEC23B only increases the level of LC3-II on autophagosomes by enhancing LC3 lipidation at the ERGIC rather than the efficiency of autophagosome formation (the extent of autophagic degradation). To confirm that FBXW5 knockdown and the expression of the SEC23(SD) mutant indeed increase the autophagic flux even under nutrient-rich conditions, the authors should determine whether degradation of autophagic cargos such as p62 is also activated.

3) Related to comment #2, if the expression of SEC23B(S186D) is indeed sufficient to induce autophagy, the data suggest that the primary function of the ULK1 complex is phosphorylation of SEC23B among many known substrates. To prove this hypothesis, it would be important to test whether the expression of SEC23B(S186D) can rescue autophagy in ULK1- or ULK1/2-double knockdown cells.

4) If the regulation of the level of SEC23B by FBXW5 is important for the control of autophagy, is overexpression of even wild-type SEC23B sufficient to induce autophagy?

5) The number of LC3 can be increased by not only autophagy activation but also inhibition of the lysosomal function. Given that overexpression of SEC23B(S186D) affects secretion (Supplementary Figure 3), the authors should rule out the possibility that it also affects the lysosomal function and thereby causes accumulation of LC3. The lysosomal function can be tested by monitoring maturation of cathepsins or degradation of the EGF receptor.

6) Subsection “Ser186 in SEC23B is necessary for its localization to the ERGIC and an efficient autophagic response upon nutrient deprivation”, first paragraph and Supplementary Figure 3D and E: The data on the secretory pathway by assaying secreted Gaussia Luciferase are critical and should be shown in the main figures rather than in the supplement. It could have been well the case that SEC23-P/SEC24A/B would lead to vesicles feeding autophagosomes *and* fueling general secretion.

7) Perhaps one important point to clarify is what happens to the secretory pathway and ER to Golgi transport with SEC23B-P/SEC24A/B? Is there hard evidence to exclude the notion of SEC23B-P/SEC24A/B during starvation does not also fuel secretion (except for the luciferase secretion that is inhibited)? Any specific cargo? Why would this complex SEC23B-P/SEC24A/B not support full COPII coat formation? The authors seem in doubt exemplified with the question mark in their model. Does SEC23-P/SEC24A/B also bind the other COPII subunits to form a full coated vesicle? This should be experimentally tested. What about the other isoforms of SEC23 and SEC24C/D?

---

## [Author Response]

Essential revisions:1) The authors propose that ULK1-mediated phosphorylation of SEC23B promotes its translocation to the ERGIC, but this is not directly shown. The authors should test whether the translocation of SEC23B to the ERGIC is indeed abolished in ULK1- or ULK1/2-knockdown cells.

We now show that ULK1 knockdown significantly inhibits the translocation of SEC23B to the ERGIC (new Figure 5B).

2) The authors conclude that FBXW5-dependent degradation of SEC23B downregulates basal autophagy based on the results of the LC3 puncta formation and mCherry-GFP-LC3 assays. However, the authors should rule out the possibility that SEC23B only increases the level of LC3-II on autophagosomes by enhancing LC3 lipidation at the ERGIC rather than the efficiency of autophagosome formation (the extent of autophagic degradation). To confirm that FBXW5 knockdown and the expression of the SEC23(SD) mutant indeed increase the autophagic flux even under nutrient-rich conditions, the authors should determine whether degradation of autophagic cargos such as p62 is also activated.

We now show that FBXW5 knockdown and the expression of the SEC23(S186D) mutant activate the degradation of p62 under nutrient-rich conditions (new Figure 4B and new Figure 4E).

3) Related to comment #2, if the expression of SEC23B(S186D) is indeed sufficient to induce autophagy, the data suggest that the primary function of the ULK1 complex is phosphorylation of SEC23B among many known substrates. To prove this hypothesis, it would be important to test whether the expression of SEC23B(S186D) can rescue autophagy in ULK1- or ULK1/2-double knockdown cells.

We now show that the expression of SEC23B(S186D) is unable to rescue autophagy in ULK1 knockdown cells (new Figure 4—figure supplement 1D). This is likely because ULK1 is necessary to phosphorylate many pro-autophagic substrates. Moreover, in presence of nutrients, the basal activity of ULK1 is necessary to promote the assembly of COPII complexes (Joo et al., 2016). Therefore, in the absence of ULK1, it is expected that neither secretory nor autophagic COPII vesicles are efficiently formed.

4) If the regulation of the level of SEC23B by FBXW5 is important for the control of autophagy, is overexpression of even wild-type SEC23B sufficient to induce autophagy?

We now show that overexpression of wild-type SEC23B to levels similar to those of the stable SEC23B(S186D) mutant is sufficient to induce autophagy (new Figure 4F).

5) The number of LC3 can be increased by not only autophagy activation but also inhibition of the lysosomal function. Given that overexpression of SEC23B(S186D) affects secretion (Supplementary Figure 3), the authors should rule out the possibility that it also affects the lysosomal function and thereby causes accumulation of LC3. The lysosomal function can be tested by monitoring maturation of cathepsins or degradation of the EGF receptor.

By monitoring the levels of p62 as well as pro-Cathepsin and active-Cathepsin, we now show that the overexpression of SEC23B(S186D) does not affect the lysosomal function (new Figure 4E and new Figure 4F).

6) Subsection “Ser186 in SEC23B is necessary for its localization to the ERGIC and an efficient autophagic response upon nutrient deprivation”, first paragraph and Supplementary Figure 3D and E: The data on the secretory pathway by assaying secreted Gaussia Luciferase are critical and should be shown in the main figures rather than in the supplement. It could have been well the case that SEC23-P/SEC24A/B would lead to vesicles feeding autophagosomes and fueling general secretion.

We have moved these figures to the main figures (now Figure 5C-D).

7) Perhaps one important point to clarify is what happens to the secretory pathway and ER to Golgi transport with SEC23B-P/SEC24A/B? Is there hard evidence to exclude the notion of SEC23B-P/SEC24A/B during starvation does not also fuel secretion (except for the luciferase secretion that is inhibited)? Any specific cargo? Why would this complex SEC23B-P/SEC24A/B not support full COPII coat formation? The authors seem in doubt exemplified with the question mark in their model. Does SEC23-P/SEC24A/B also bind the other COPII subunits to form a full coated vesicle? This should be experimentally tested. What about the other isoforms of SEC23 and SEC24C/D?

The phosphorylation of Ser186 does not increase the affinity of SEC23B to SEC24A and SEC24B. Rather, it inhibits the binding of SEC23B to SEC24C and SEC24D. We never claimed in the text that the binding of phosphorylated SEC23B to SEC24A and SEC24B reduces secretion. The lack of binding between phosphorylated SEC23B and SEC24C/D is striking and we speculate that this inhibition in binding may explain the reduced secretion. However, we have shown that SEC23B(S186D) (which mimics the phosphorylated form of SEC23B) binds to SAR1(H79G), a GTP-bound SAR1 mutant that is constitutively associated with COPII vesicles, but not with the cytosolic GDP-bound SAR1(T39N) mutant (Figure 3—figure supplement 1E). Moreover, SEC23B(S186D) also interacts with SEC13, SEC16, and SEC31 (new Figure 6A). In conclusion, we believe that, upon starvation, SEC23B-SEC24A/B promotes autophagy, but can possibly also sustain secretion since these proteins can bind other COPII subunits to form a full-coated vesicle. However, the decrease in the abundance of the SEC23B-SEC24C/D complexes likely contributes to a decrease in secretion. Concerning the other isoform of SEC23, Gan et al. (2017) have shown that ULK1 induces the dissociation of SEC23A from SEC31A in response to starvation, further inhibiting the secretory pathway. We now integrated better these points on the Discussion section. We also agree that the question mark going from the SEC23B-SEC24A/B to the “ER-to-Golgi trafficking”, which we drew in the old cartoon (old Supplementary Figure 4E) is confusing and reductionist. In fact, we believe that a simple cartoon cannot easily summarize the effects of FBXW5, ULK1, SEC23B phosphorylation, and SEC23B accumulation on both secretion and autophagy. Therefore, we eliminated the cartoon and its shortcomings from the supplementary figures. Finally, we believe that the identification of specific cargos goes beyond the scope of the current study.